# Persistent, Extensive Channelized Drainage Modeled Beneath Thwaites Glacier, West Antarctica

Alexander O. Hager[1,2], Matthew J. Hoffman[2], Stephen F. Price[2], and Dustin M. Schroeder[3,4]

[1]Department of Earth Sciences, University of Oregon, Eugene, OR, USA
[2]Fluid Dynamics and Solid Mechanics Group, Los Alamos National Laboratory, Los Alamos, NM, USA
[3]Department of Electrical Engineering, Stanford University, Stanford, CA, USA
[4]Department of Geophysics, Stanford University, Stanford, CA, USA

**Correspondence:** Alexander Hager (ahager@uoregon.edu)

**Abstract.** Subglacial hydrology is a leading control on basal friction and the dynamics of glaciers and ice sheets. At low discharge, subglacial water flows through high-pressure, sheet-like systems that lead to low effective pressures. However, at high discharge, subglacial water melts the overlying ice into localized channels that efficiently remove water from the bed, thereby increasing effective pressure and basal friction. Recent observations suggest channelized subglacial flow exists beneath Thwaites Glacier, yet it remains unclear if stable channelization is feasible in West Antarctica, where surface melting is nonexistent and water at the bed is limited. Here, we use the MPAS-Albany Land Ice model to run a suite of over 130 subglacial hydrology simulations of Thwaites Glacier across a wide range of physical parameter choices to assess the likelihood of channelization. We then narrow our range of viable simulations by comparing modeled water thicknesses to previously observed radar specularity content, which indicates flat, spatially extensive water bodies at the bed. In all of our data-compatible simulations, stable channels reliably form within 100–200 km of the grounding line, and reach individual discharge rates of 35–110 $m^3$ $s^{-1}$ at the ice-ocean boundary. While only one to two channels typically form across the 200 km width of the glacier in our simulations, their high efficiency drains water across the entire lateral extent of the glacier. We posit the large catchment size of Thwaites Glacier, its funnel-like geometry, and high basal melt rates together accumulate enough water to form stable channels. No simulations resembled observed specularity content when channelization is disabled. Our results suggest channelized subglacial hydrology has two consequences for Thwaites Glacier dynamics: (i) amplifying submarine melting of the terminus and ice shelf, while (ii) simultaneously raising effective pressure within 100 km of the grounding line and increasing basal friction. The distribution of effective pressure implied from our modeling differs from parameterizations typically used in large-scale ice sheet models, suggesting the development of more process-based parameterizations may be necessary.

## 1 Introduction

Subglacial hydrology is a leading control on basal friction and frontal ablation rates of tidewater glacier termini, yet the morphology of subglacial drainage systems beneath the Antarctic Ice Sheet poorly characterized. Subglacial water can either flow through a highly pressurized, distributed network of bedrock cavities (Walder, 1986; Kamb, 1987), sediment canals

(Walder and Fowler, 1994), films (Weertman, 1972), and porous till (Clarke, 1987), or efficiently drain through arborescent channels melted upward into basal ice (Röthlisberger, 1972). Water flow through a distributed system creates low effective pressures contributing to fast basal sliding (Walder, 1986; Kamb, 1987), whereas channelized drainage increases effective pressures (Röthlisberger, 1972; Schoof, 2010; Hewitt, 2011) and local submarine melt rates at the ice-ocean boundary (Slater et al., 2015). To date, most models of basin or ice sheet-scale Antarctic subglacial drainage have focused on hydropotential mapping (e.g., Stearns et al., 2008; Carter and Fricker, 2012; Le Brocq et al., 2013; Livingstone et al., 2013; Smith et al., 2017), and have only recently distinguished between conduit types under Antarctic glaciers (Dow et al., 2020; Wei et al., 2020). However, a growing body of work suggests a variety of drainage styles may be important in Antarctica, with obvious relevance to ice sheet dynamics.

In Antarctica, shallow hydropotential gradients and the lack of significant surface melt has led to the conventional paradigm that subglacial water fluxes are too small to permit stable channelized drainage beneath the ice sheets (e.g., Weertman, 1972; Alley, 1989; Walder and Fowler, 1994; Carter et al., 2017). This assumption has led to the use of purely distributed subglacial hydrology models (e.g., Alley, 1996; Le Brocq et al., 2009), or simplifying approximations of effective pressure in large-scale Antarctic ice sheet models (e.g., Leguy et al., 2014; Asay-Davis et al., 2016; Yu et al., 2018; Nias et al., 2018; Cornford et al., 2020). However, channelized drainage under Antarctic ice sheets has recently been inferred through observations of ice shelf basal melt channels (Le Brocq et al., 2013; Marsh et al., 2016; Drews et al., 2017), radar specularity content (Schroeder et al., 2013), and subglacial hydrology models (Dow et al., 2020; Wei et al., 2020). In the absence of surface meltwater, subglacial channels must be sustained through basal melting, and the presence of basal melt channels under ice shelves suggest that their grounded counterparts must persist stably for decades or centuries (Le Brocq et al., 2013; Marsh et al., 2016).

Thwaites Glacier contains enough ice to raise sea level 65 cm (Rignot et al., 2019), and may currently be undergoing an unstable retreat, likely triggered by increased melting of its ice shelf and terminus (Joughin et al., 2014; Rignot et al., 2014; Seroussi et al., 2017; Milillo et al., 2019; Hoffman et al., 2019). Ice flux from Thwaites Glacier increased 76% between 1976–2013 (Mouginot et al., 2014), coinciding with thinning rates of up to $10 \text{ m yr}^{-1}$ and a surface acceleration of $100 \text{ m yr}^{-1}$ near the grounding line (Pritchard et al., 2009; Helm et al., 2014; Gardner et al., 2018). While bed topography primarily regulates Thwaites Glacier retreat, uncertainty in basal friction laws, ice flow models, and ice shelf melt parameterizations could affect mass loss projections for this century by up to 300% (Yu et al., 2018). As a prominent control on both basal friction and submarine melting, subglacial hydrology has the potential to be a critical component of Thwaites Glacier dynamics, yet the configuration of its drainage network is poorly understood.

Using a recent survey of radar specularity content, Schroeder et al. (2013) hypothesized that channelized subglacial drainage is pervasive within $75 - 100 \text{ km}$ of the Thwaites Glacier grounding line. However, subsequent satellite detection of subglacial lakes led to the interpretation that such channels may only be ephemeral, forming only during lake drainage events (Smith et al., 2017). Here, we pair remote sensing with the 2-dimensional subglacial hydrology model implemented within the MPAS-Albany Land Ice Model (MALI) (Hoffman et al., 2018), to provide a more complete picture of the likely configuration of the Thwaites Glacier subglacial drainage system. We run a suite of 138 modeling simulations, then compare our results with the observed radar specularity content of Schroeder et al. (2013) to define a subset of scenarios as possible representations of

reality. Results from this subset are then collated with ice shelf basal melt rates and common parameterizations of basal friction to explore the significance of channelization on submarine melt rates and ice dynamics.

## 2    Methods

### 2.1    Model Framework

Here, we use only the subglacial hydrology component of MALI, which contains both distributed and channelized flow components, and operates on an unstructured, two-dimensional Voronoi grid. Velocities and fluxes are calculated on the edge midpoints of each cell, and all other variables are located at cell centers. Channel segments connect the centers of neighboring cells. The distributed system is treated as a macroporous sheet that is designed to resemble the bulk flow of water through cavities on the lee-sides of bedrock bumps (Flowers and Clarke, 2002; Hewitt, 2011; Flowers, 2015), but may also reasonably describe flow through other porous media, such as till or till canals (Hewitt, 2011; Flowers, 2015; Hoffman et al., 2016). The distributed system discharge is given by:

$$\boldsymbol{q} = -k_q W^{\alpha_1} \left| \nabla \phi \right|^{\alpha_2} \nabla \phi \tag{1}$$

where $k_q$ is the conductivity coefficient of the distributed system, $W$ is the water thickness, and $\alpha_1$ and $\alpha_2$ are $\frac{5}{4}$ and $-\frac{1}{2}$, respectively, to resemble a Darcy-Weisbach flow law. The hydropotential, $\phi$, is defined as:

$$\phi = \rho_w g Z_b + P_w. \tag{2}$$

where $\rho_w$ is the water density, $g$ is the gravitational acceleration, $Z_b$ is the bed topography (Figure 1a), and $P_w$ is the distributed water pressure. It is assumed all basal cavities remain filled, and thus water thickness is a function of cavity opening from basal sliding over bedrock bumps and creep closure:

$$\frac{dW}{dt} = c_s \left| \boldsymbol{u}_b \right| (W_r - W) - c_{cd} A_b N^3 W \tag{3}$$

where $c_s$ is a bed roughness parameter, $\boldsymbol{u}_b$ is the ice basal sliding velocity (Figure 1b), $W_r$ is the maximum bed bump height, $c_{cd}$ is a creep scaling parameter for the distributed system, and $A_b$ is the temperature-dependent ice flow rate parameter for basal ice. The effective pressure, $N$, is defined as the difference between the ice overburden and water pressures: $N = \rho_i g H - P_w$, for ice thickness $H$ and ice density $\rho_i$.

The channelized system formulation resembles that of Werder et al. (2013), where channel discharge is given by:

$$\boldsymbol{Q} = -k_Q S^{\alpha_1} \left| \nabla \phi \right|^{\alpha_2} \nabla \phi \tag{4}$$

where $k_Q$ is the channel conductivity coefficient. Channel cross-sectional area, $S$, is a function of creep closure, and melting/freezing due to the dissipation of potential energy, $\Xi$, and pressure-dependent changes to the sensible heat of water, $\Pi$:

$$\frac{dS}{dt} = \frac{1}{\rho_i L} (\Xi - \Pi) - C_{cc} A_b N^3 S. \tag{5}$$

Here, $L$ is the latent heat of melting and $C_{cc}$ is a creep scaling parameter for channels. $\Xi$ includes dissipation terms for both the distributed and channelized systems, so that:

$$\Xi = \left| \boldsymbol{Q}\frac{d\phi}{ds} \right| + \left| l_c q_c \frac{d\phi}{ds} \right| \tag{6}$$

where $s$ is the along-channel spatial coordinate, and $q_c$ is the discharge in the distributed system within a distance, $l_c$, from the channel. Using this formulation, channels may only develop if there exists sufficient discharge in the distributed system for melting to overcome creep closure. In our experiments, we disabled the pressure-dependent melting/freezing term, $\Pi$, to avoid nonphysical instabilities arising from intricate bed topography. The implications of neglecting this term are discussed in Section 4.3.

Closing the system of equations requires the conservation of water mass within the combined distributed and channelized subglacial drainage systems, and a conservation of energy equation for the production of basal meltwater. Conservation of mass is written as:

$$\frac{dW}{dt} = -\nabla \cdot \boldsymbol{q} - \left[ \frac{\partial S}{\partial t} + \frac{\partial Q}{\partial s} \right] \delta(x_c) + \frac{m_b}{\rho_w}, \tag{7}$$

where $\delta(x_c)$ is the Dirac delta function applied along the locations of the linear channels and $m_b$ is the production of basal meltwater (Figure 1d). Conservation of energy is written as

$$m_b L = G + \boldsymbol{u}_b \cdot \boldsymbol{\tau_b} \tag{8}$$

for basal shear $\boldsymbol{\tau_b}$ and geothermal flux $G$.

Time derivatives are discretized using an explicit forward Eulerian method that fulfills advective and diffusive Courant-Friedrichs-Lewy (CFL) conditions for the distributed system, and advective CFL conditions for the channelized system. Model outputs are written at one month intervals and all reported results are averaged over five years of model time to smooth any minor oscillations remaining in the system.

## 2.2 Thwaites Model Domain

We ran the majority of our simulations on a variable resolution domain of Thwaites Glacier that has a 4 km cell spacing over the fast flowing regions and coarsens to 14 km at the interior ice divide, for a total of 4267 grid cells. An additional simulation was performed with a higher resolution mesh that uses 1 km cell spacing in fast flowing regions, coarsening to 8 km at the interior ice divide, for a total of 75500 cells. The bedrock and ice geometry were interpolated onto the model mesh using conservative remapping from the BedMachine Antarctica v1 ice thickness and bed elevation dataset (Morlighem et al., 2020). However, a maximum bed elevation of 1200 m and a ice thickness of 550 m was imposed over Mt. Takahe (>250 km from the terminus) to avoid instabilities arising from steep bed topography. The resulting thickness gradients were then smoothed by running only the ice dynamics and geometry evolution portions of MALI for 15 years. The geothermal flux was interpolated from the 15 km resolution dataset of Martos et al. (2017). The ice sliding velocity ($\boldsymbol{u}_b$) and basal shear stress ($\boldsymbol{\tau}_b$) fields required by the subglacial hydrology model follow the methods used by Hoffman et al. (2018) to generate a present-day initial

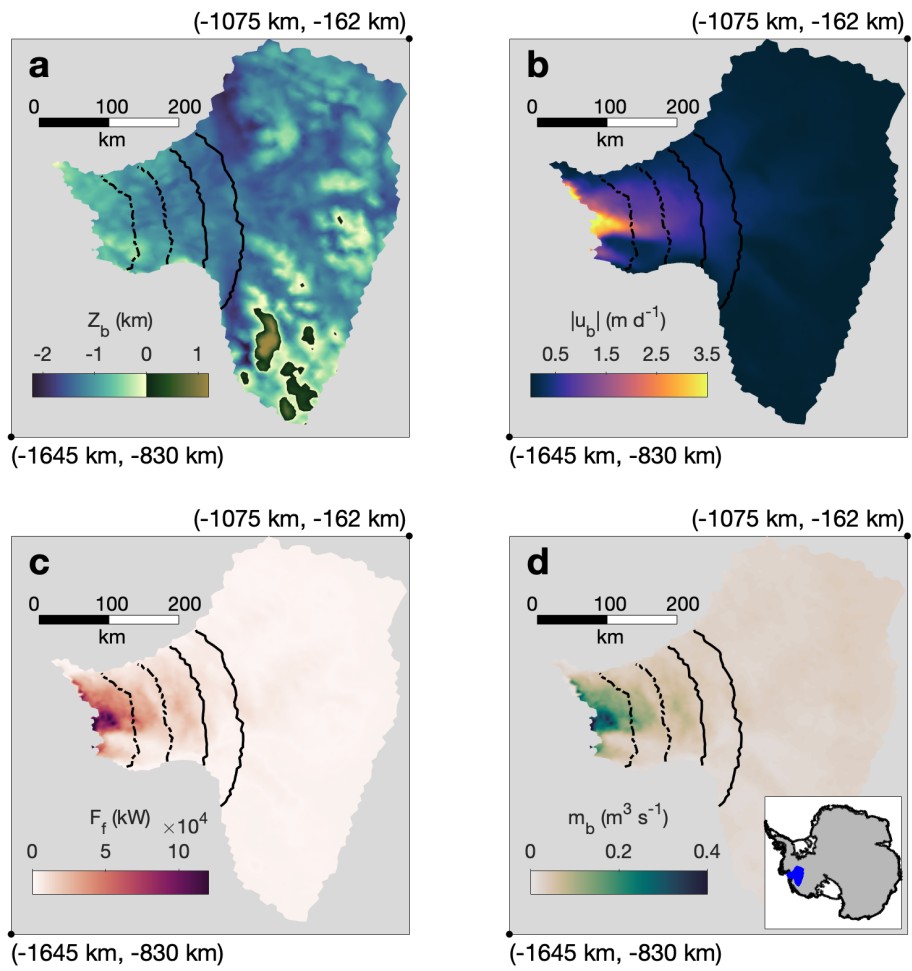

**Figure 1. a)** Bed topography ($Z_b$), **b)** basal sliding speed ($|\boldsymbol{u}_b|$), **c)** basal friction heat flux ($F_f$), and **d)** the production of basal meltwater ($m_b$) used as inputs for the subglacial hydrology model. Transects spaced every 50 km from the terminus (used for determination of flux steady-state and in Figure 6) are shown as black lines, with the dotted lines spanning the transition zone of Schroeder et al. (2013). The locations of map corners are given in Standard Antarctic Polar Stereographic coordinates. The inset in **d** depicts the location of Thwaites Glacier (blue) within Antarctica.

condition, where a basal friction parameter is optimized in order to minimize the misfit between modeled and observed ice surface velocity (Perego et al., 2014).

Within the subglacial hydrology model, no flow lateral boundary conditions were applied at the ice-covered lateral boundaries of the model domain. At the glacier grounding line, a Dirichlet boundary condition on the hydropotential ($\phi$) was applied equal to the hydropotential of seawater at each grid cell seaward of the grounding line,

$$\phi_o = \rho_w g Z_b - \rho_o g Z_b, \tag{9}$$

where $\rho_o = 1028$ kg m$^{-3}$ is the density of ocean water. Note this boundary condition results in hydropotential values close to zero but spatially varying as ocean pressure varies along the grounding line with the thickness of the ocean water column. Additionally, inflow from the ocean to the subglacial drainage system is disallowed if the hydropotential underneath the grounded ice falls below the ocean hydropotential. This condition can occur due to a spatially variable, ocean-lateral boundary condition and the assumption of constant density within the subglacial drainage system, which in combination with subglacial channelization, can locally result in the modeled unstable inflow of ocean water. The model was spun-up with channelization disabled and a $k_q$ value of $1.5 \times 10^{-3}$ m$^{7/4}$ kg$^{-1/2}$ to allow water pressures to equilibrate at $> 90\%$ overburden pressure. All other simulations were then initialized from the steady-state solution of this run.

## 2.3 Parameter Sweep and Sensitivity Analysis

Four primary yet poorly constrained parameters exist in equations 1, 3, and 4: $k_q$, $k_Q$, $W_r$, and $c_s$. While some theoretical and observational basis exists for the values of these parameters, the appropriate values are uncertain and likely vary by glacier basin. A few recent studies have addressed this uncertainty by using inversion techniques to infer values of hydraulic parameters (e.g., Brinkerhoff et al., 2016; Koziol and Arnold, 2017; Irarrazaval et al., 2021; Brinkerhoff et al., 2021). Here, we used an ensemble approach and compared results to multiple limiting criteria to identify the most realistic parameter combinations. Our ensemble consisted of 113 different channel-enabled simulations and 25 simulations disallowing channelization. All runs were within a plausible parameter space based on observations and theory, as described below.

Observations of jökulhlaups suggest the typical Manning roughness, $n$, of subglacial channels ranges from $0.023 - 0.12$ m$^{-1/3}$ s (Nye, 1976; Clarke, 1982; Bjornsson, 1992; Clarke, 2003). We can translate these Manning roughness values to the equivalent channel conductivity range of $0.03 - 0.17$ m$^{7/4}$ kg$^{-1/2}$ using (Werder et al., 2013):

$$k_Q{}^2 = \frac{1}{\rho_w g n^2 (\frac{2}{\pi})^{2/3} (\pi + 2)^{4/3}}. \tag{10}$$

However, jökulhlaups do not provide an exhaustive range of roughness characteristics for channel flow, and dye-trace breakthrough curves have indicated that $n$ values for low-discharge, high-friction subglacial channels could be as low as $n = 0.68$ m$^{-1/3}$ s (Gulley et al., 2012), or $k_Q = 0.006$ m$^{7/4}$ kg$^{-1/2}$. On the other extreme, the Manning roughness of a smooth brass pipe is $0.009$ m$^{-1/3}$ s (Chow, 1959), or $k_Q = 0.44$ m$^{7/4}$ kg$^{-1/2}$, which we consider a generous upper end-member for $k_Q$. We therefore ran our model with $k_Q$ ranging from $0.005 - 0.5$ m$^{7/4}$ kg$^{-1/2}$ to encompass the full set of plausible values.

Because $k_q$ may be chosen to portray porous flow through cavities in till or bedrock, we selected $k_q$ values to be within the appropriate range of till or greater. Estimates for the hydraulic conductivity, $\kappa$, of subglacial till ranges widely from $10^{-12} - 5 \times 10^{-4}$ m s$^{-1}$ (Fountain and Walder, 1998), which can be converted to an equivalent distributed conductivity coefficient in our model via (Bueler and van Pelt, 2015):

$$k_q = \frac{\kappa}{\rho_w g W^{1/4} |\nabla \phi|^{-1/2}}. \tag{11}$$

Using a characteristic $W$ of 0.1 (see below) and $|\nabla \phi|$ of 100 Pa m$^{-1}$ (approximated from our model domain), we estimate the conductivity coefficient of subglacial till in our model would be $10^{-15} - 10^{-6}$ m$^{7/4}$ kg$^{-1/2}$, which should span our lower

limit for $k_q$. In practice, however, simulations with $k_q < 1.5 \times 10^{-5}$ m$^{7/4}$ kg$^{-1/2}$ were over-pressurized and did not reach steady-state. Although no proper upper bound exists for $k_q$, we attempted to limit our $k_q$ parameter sweep to values that kept the average water pressure $> 90\%$ flotation, which typically occurred for $k_q \leq 5 \times 10^{-3}$ m$^{7/4}$ kg$^{-1/2}$ across different bed
roughness combinations. This choice was based off of near flotation water pressures observed at Ice Stream B (Engelhardt and Kamb, 1997) and estimated for Pine Island Glacier (Gillet-Chaulet et al., 2016), which we assume are similar to those beneath Thwaites Glacier.

In theory, $W_r$ represents the characteristic bed bump height (decimeter-scale), while $c_s$ represents the characteristic meter-scale bed bump spacing (Figure 2). Typical values used for $W_r$ and $c_s$ are $\sim 0.1$ m (e.g., Schoof, 2010; Hewitt, 2011; Schoof
et al., 2012; Werder et al., 2013; de Fleurian et al., 2018; Dow et al., 2020) and $\sim 0.5$ m$^{-1}$ (e.g., Schoof et al., 2012; Werder et al., 2013; Hoffman and Price, 2014; de Fleurian et al., 2018; Dow et al., 2020), respectively. We tested the sensitivity of our results to these parameters by running the model with 6 different combinations of $W_r = 0.05$ m, 0.1 m, 0.2 m, and 1.0 m and $c_s = 0.25$ m$^{-1}$, 0.5 m$^{-1}$, and 1.0 m$^{-1}$, holding one at the default value of $W_r = 0.1$ m or $c_s = 0.5$ m$^{-1}$, and varying the other parameter. We spaced $k_q$ and $k_Q$ samples at consistent intervals, and stopped sampling conductivity parameter space when
runs failed to reach steady-state or were under-pressurized ($< 90\%$ flotation). As a result, we conducted a different number of runs for each bed parameter combination, ranging between $9 - 29$ channel-enabled simulations with $k_q$ and $k_Q$ values within their plausible ranges (Appendix A).

Additionally, for each pair of bed roughness parameters, we ran 4-5 simulations with channelization disabled across a similar range of $k_q$ values (25 runs total). These were used as counter-examples to explore the impact of subglacial channel drainage
under Thwaites Glacier.

By design, the parameter sweep forces our model to operate at the limit of its ability to remain stable, and thus some runs failed to reach a true steady-state. This occurred for two main reasons: either local numerical instabilities developed in the channel model, or the domain became over-pressurized so that the adaptive timestep became impractically small to meet the pressure CFL condition. We thus found it useful to define two separate steady-state criteria that allowed us to identify which
information was usable from each run, and categorized runs as either reaching a pressure steady-state or a flux steady-state. Pressure steady-state was defined as $\langle \frac{\partial N_{ij}}{\partial t} N_{ij}^{-1} \rangle \leq 0.5\%$, where $\langle \rangle$ denotes an average over all grounded grid cells $j$ and time steps $i$ over 5 years of model time. Flux steady-state was attained when the area-integrated version of equation 7 upstream of a specified cross-glacier transect was met within $0.5\%$ when averaged over 5 years. Transects were defined every 50 km within 200 km of the grounding line (Figure 1). Runs that failed to reach flux steady-state did not represent steady systems
where the subglacial discharge realistically balanced the production of meltwater, and so it was not possible to accurately assess the relative fraction of channel discharge to distributed system discharge. Therefore, we report results regarding water thickness and water pressure from pressure steady-state runs, but only report discharge results from runs that also reached flux steady-state at each transect.

We use this approach because water pressure and thickness fields from pressure steady-state runs strongly resemble their
flux steady-state neighbors in parameter space, yet the channel model fails to reach equilibrium in some runs due to local channel instabilities that do not affect area-averaged water pressure or water thickness. We thus have confidence that pressure

steady-state runs still yield useful information about water pressure and thickness. In some cases, instabilities could be avoided by changing the englacial porosity, which acts as a buffer between meltwater production and the subglacial system but does not affect the steady-state configuration. As our goal was to explore as much of parameter space as possible, runs were continually restarted until either reaching flux steady-state, forming an unpreventable numerical instability, or becoming computationally untenable to keep running. Simulations that did not reach either steady-state criteria were discarded. The sensitivity of our results to our steady-state criteria is discussed in Appendix A.

## 2.4 Model Comparison with Observed Specularity Content

All simulations that reached a pressure steady-state were compared with observed radar specularity content from Thwaites Glacier (Schroeder et al., 2013) to further narrow the range of viable parameter combinations. Specularity content determined from airborne ice-penetrating radar is commonly used for detecting subglacial water bodies beneath ice sheets (e.g., Schroeder et al., 2013, 2015; Young et al., 2016, 2017; Dow et al., 2020), and has recently been used to validate a subglacial hydrology model of Totten Glacier, East Antarctica (Dow et al., 2020). Although our methods differ, we rely on the same concepts that make specularity content a useful tool for subglacial hydrology model validation.

Ponding within the subglacial drainage system creates flat, reflective surfaces that cause bright specular returns, as opposed to bedrock, which has a lower dielectric contrast to ice, and whose rough texture scatters energy (Schroeder et al., 2015). Similarly, the curved surface of less uniform conduits such as channels or rough linked cavities scatters energy uniformly in all directions, creating areas of low specularity content, despite the presence of water (Schroeder et al., 2013). High specularity content, therefore, unequivocally depicts flat-surfaced water bodies in an inefficient distributed system, while low specularity content can either represent a distributed system below its capacity (bedrock cavities are smaller than their maximum size allowed by bed roughness), or the existence of water in rougher, more variably shaped conduits, such as channels. However, by comparing specularity content with a numerical model, we are able to determine which of these two features is responsible for creating the weakly specular regions beneath Thwaites Glacier.

To compare specularity content with our model output, we first averaged the specularity content from the North-South and East-West radar transects from Schroeder et al. (2013) onto a 5 km grid. We then defined a water thickness to bump height ratio, $R_{wt}$, which indicates the degree to which modeled conditions would produce flat and extensive interfaces between water and ice at the glacier bed, and therefore highly specular surfaces:

$$R_{wt} = \frac{W}{W_r}. \tag{12}$$

For $R_{wt} \gtrsim 1$, distributed water thickness nears or exceeds bed bump height, thus creating a flat, highly-specular surface of water. However, for $R_{wt} \ll 1$ bedrock geometry determines the roughness of the lower interface, and the location is considered rough-surfaced and non-specular (Figure 2). Additionally, with a proper value of $k_q$, $R_{wt}$ can also parameterize till saturation, with low and high $R_{wt}$ indicating under-saturated (non-specular) till and saturated (specular) till, respectively. For easy comparison, $R_{wt}$ was calculated for each model grid cell, then interpolated onto the same 5 km grid as the specularity content data. Note that a spatially uniform $W_r$ is likely unrealistic but is an assumption commonly used in subglacial hydrology models.

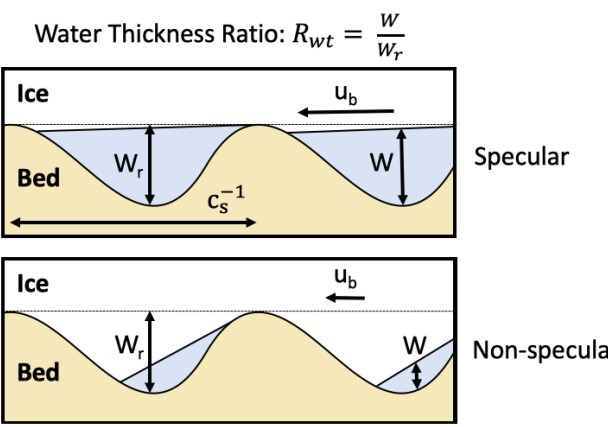

**Figure 2.** Schematic of a specular and non-specular distributed system, as defined by the water thickness ratio, $R_{wt}$. Physical representations of bed roughness parameters are included.

As applied here, Equation 12 is used as a relative metric of how close to maximum size a linked cavity system is, and this interpretation would apply to both uniform or spatially variable bump heights.

Measured specularity content and modeled $R_{wt}$ both represent broad, flat areas of pooled water at high values, but should not be expected to covary when their values are low, due to nonlinearities in the measurements and model formulations, as well as ambiguity in the physical representation of low specularity content. This makes comparing the two difficult, and a simple

spatial correlation unlikely to work as a comparison method. Instead, we rely on binary masks that map where specularity content and $R_{wt}$ are high/low, as determined by their value being above/below a threshold value. Unfortunately, this method requires choosing thresholds for what is considered high for each quantity, which we address by creating a population of masks for each variable, each using a different threshold within a reasonable range.

Specularity content depends on the geometry of ice thickness, survey geometry, radar processing, and subglacial water

geometry (Schroeder et al., 2013, 2015; Young et al., 2016; Haynes et al., 2018). As a result, specularity content can be interpreted as the relative amount of the bed that is covered by flat subglacial water bodies, which gradually transitions from non-specular to specular with the addition of water. Therefore, the classification of high or low specularity content is determined relative to a specific survey, and we base the threshold value used for creating specularity masks on the cumulative distribution of specularity within our dataset (Figure B2). As masks are sensitive to the exact choice of threshold, we created 11 specularity

masks with thresholds, $S^{crt}$, ranging from $0.15 - 0.25$ at evenly-spaced intervals of 0.01, which selects for the greatest $\sim$ $5 - 20\%$ of our specularity data. Similarly, we assume specular surfaces require cavities that are near maximum size (Figure 2), so there must be a range of $R_{wt}$ near 1 that could plausibly represent high specularity. Again, we account for this range by creating 6 masks of $R_{wt}$ using thresholds, $R_{wt}^{crt}$, between $0.95 - 1.0$ at intervals of 0.01. The resultant 66 combinations of specularity content and $R_{wt}$ masks were then compared using two criteria:

1. The masks were divided into four zones based on Schroeder et al. (2013): a near-terminus non-specular zone thought to have channelized flow, a lower specular zone approximately at the transition zone of Schroeder et al. (2013), an upper specular zone where ponding is thought to occur, and an upper non-specular zone likely containing little basal water (Figure 3). The specularity content and $R_{wt}$ masks had to agree for a majority of the cells within each zone.

2. The two masks needed to have an overall correlation coefficient of $\geq 0.35$, which was empirically tuned to select for similar patterns between masks when paired with the first criterion.

Model runs that had at least one $R_{wt}$ mask meet these comparison criteria with at least one specularity mask were deemed data-compatible and used for further analysis. By admitting runs that satisfy the comparison criteria for even a single set of masks out of the 66 compared, we make the selection highly inclusive so that conclusions about extent of channelization consider the widest range of parameters compatible with specularity observations. Hereafter, runs that additionally met flux steady-state criteria will be referred to as data-compatible FSS runs. See Appendix B for more information about these comparison criteria, as well as a flow chart illustrating the comparison process (Figure B1).

## 3 Results

### 3.1 Channel-Enabled Parameter Sweep

#### 3.1.1 Model Tuning and Correspondence with Specularity Content

Of our 113 channel-enabled runs, 39 met our pressure steady-state criterion, while 23 of those also met our flux steady-state criterion across all transects. 20 pressure steady-state runs, including 13 flux steady-state runs, had at least one $R_{wt}$ and specularity mask combination that met our comparison criteria, and were therefore considered possible representations of reality. Each of these runs showed a strong resemblance between $R_{wt}$ and specularity content masks (Figure 3). Average water pressures in data-compatible runs were between 91-96% flotation, and in general, runs that did not correspond with specularity content had water pressures outside of this range.

All 66 combinations of $S^{crt}$ and $R^{crt}_{wt}$ masks yielded successful comparisons for some sets of parameters, although successful pairings varied with model parameters. Across all runs, comparison success rate exponentially increased with higher values of $R^{crt}_{wt}$, with $R^{crt}_{wt}$ of 0.99 or 1.0 accounting for 60% of all matches. Conversely, masks with $R^{crt}_{wt} = 0.95$ only accounted for 4% of the 713 successful mask combinations. The few runs that had successful matches with an $R^{crt}_{wt}$ of 0.95 also had successful matches using higher $R^{crt}_{wt}$ thresholds, indicating this choice of lower bound does not influence our results. Match success rate was not sensitive to $S^{crt}$, and each threshold value was responsible for 7–10% of successful matches.

Data-compatible runs either had $k_q$ values of $1.5 \times 10^{-4}$ or $5 \times 10^{-4}$ m$^{7/4}$ kg$^{-1/2}$ (Figure 4), with the only exceptions occurring when $W_r = 0.05$ m or $W_r = 1.0$ m, in which data-compatible $k_q$ values reached $1.5 \times 10^{-3}$ and $5 \times 10^{-5}$ m$^{7/4}$ kg$^{-1/2}$, respectively. The range of $k_q$ in data-compatible runs is above that of pure glacial till, and is consistent with a bed composed of both till and bedrock, as is thought to be the case for Thwaites Glacier (Joughin et al., 2009; Muto et al., 2019b, a).

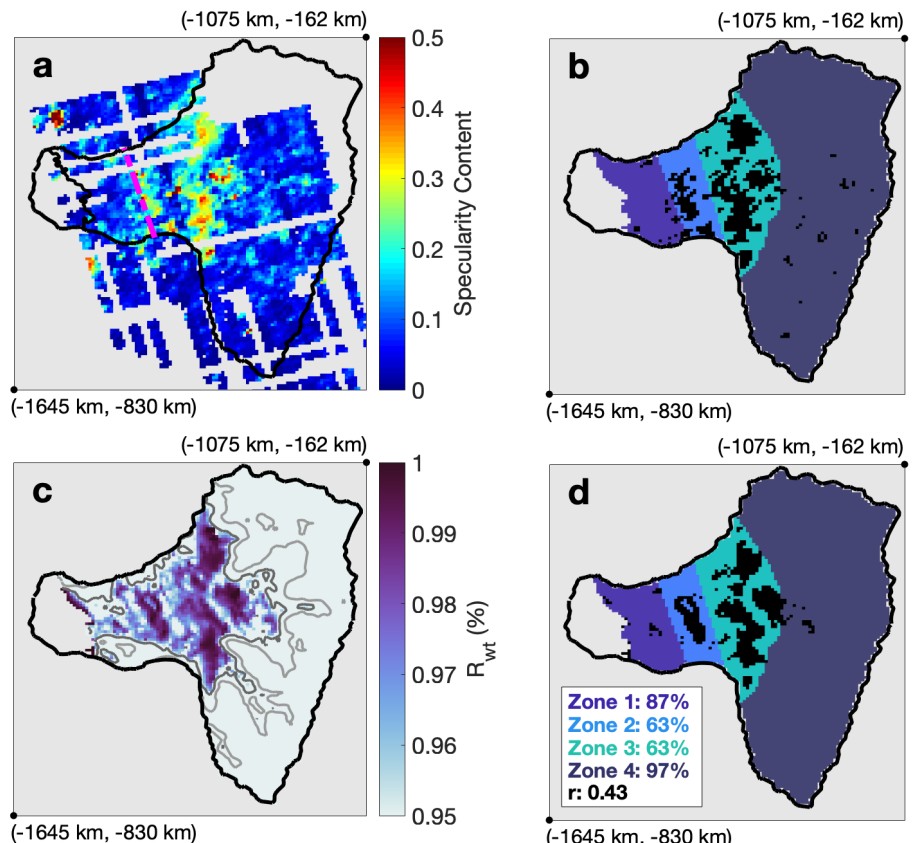

**Figure 3.** An example comparison of catchment-scale features identified with binary masks (black) of observed specularity content and modeled $R_{wt}$. **a)** radar specularity content (Schroeder et al., 2013) and **c)** $R_{wt}$ for a data-compatible flux steady-state model run, together with their coinciding binary masks, **b)** ($S^{crt} = 0.19$) and **d)** ($R_{wt}^{crt} = 0.98$), respectively. The pink dashed line in **a** marks the transition between highly-specular, distributed drainage and channel-dominated drainage, as hypothesized in Schroeder et al. (2013). The four zones used for comparison between specularity content and $R_{wt}$ are color-coded in **b** and **d**. Light and dark gray lines in **c** are the 50% and 90% $R_{wt}$ contours, respectively. The percent match between masks within each zone and the overall correlation are given in **d**. The locations of map corners are given in Standard Antarctic Polar Stereographic coordinates.

For the channelized conductivity values, all data-compatible runs had $k_Q$ values of $0.005 - 0.1$ m$^{7/4}$ kg$^{-1/2}$, coinciding with the expected range given by dye-trace breakthrough curves and Jökulhlaup observations (Nye, 1976; Clarke, 1982; Bjornsson, 1992; Clarke, 2003; Gulley et al., 2012). No runs with $k_Q = 0.5$ m$^{7/4}$ kg$^{-1/2}$, outside of our brass pipe upper limit, reached either steady-state criterion. Typical channel velocities in our data-compatible runs do not exceed the typical observed Jökulhlaup range of $0.6 - 2.7$ m s$^{-1}$ (Magnusson et al., 2007; Werder and Funk, 2009, Figure 4), which provides an additional loose constraint on the validity of our channel model, although currently no observations of subglacial flow velocities exist from Antarctica.

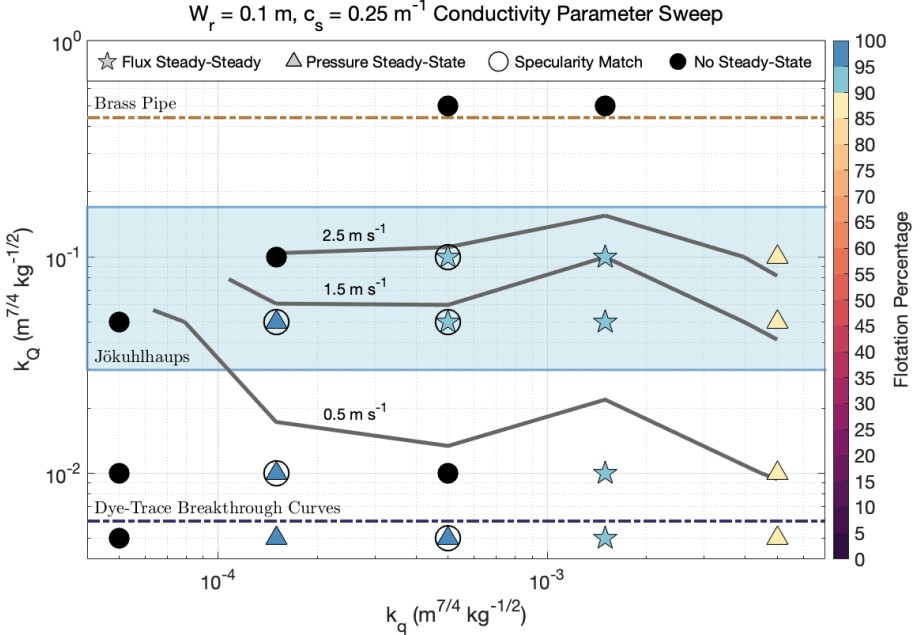

**Figure 4.** The conductivity parameter sweep for bed roughness parameters $W_r = 0.1$ m and $c_s = 0.25$ m$^{-1}$. Stars represent runs that reached flux (and pressure) steady-state, triangles symbolize pressure steady-state simulations, and filled black circles depict runs that did not reach either steady-state criterion. Symbols for steady-state runs are color-coded by the average flotation percentage of grounded ice. Circles around stars or triangles indicate runs that matched observed specularity content, and are considered data-compatible. Gray lines are $95^{th}$ percentile channel velocity contours for channels with $Q > 5$ m$^3$ s$^{-1}$. $k_Q$ limits determined from a brass pipe and dye-trace breakthrough curves are plotted as brown and dark-blue dashed lines, respectively, and the blue shaded area represents the typical observed Jökulhlaup $k_Q$ range.

### 3.1.2 Extent of Channelization in Data-Compatible Simulations

Subglacial channels were ubiquitous in all data-compatible FSS runs. In most of these runs, channels with discharges over 5 m$^3$ s$^{-1}$ extended at least 150 km from the glacier terminus, with some channels reaching farther than 200 km (Figure 5). The initiation of these channels generally coincided with the upper specular zone observed in Schroeder et al. (2013). However, channel discharge between 150 - 200 km was divided between 2 to 4 small channels, each with an individual discharge of less than 20 m$^3$ s$^{-1}$. At 150 km from the terminus, distributed discharge was still the dominant mode of drainage, with average channelized and distributed discharges of $27 \pm 18$ and $42 \pm 19$ m$^3$ s$^{-1}$ ($\pm$ indicates standard deviations), respectively, across data-compatible runs.

A transition occurs between 50–100 km from the terminus from a distributed-dominated to a channel-dominated system, coinciding with the region where Schroeder et al. (2013) hypothesized channelization begins under Thwaites Glacier. In our model, all data-compatible runs had formed at least one channel transporting $> 10$ m$^3$ s$^{-1}$ by 100 km from the terminus, and by 50 km, these channels had grown and converged into 1-2 primary channels, each draining up to 50 m$^3$ s$^{-1}$ of water. Our 50

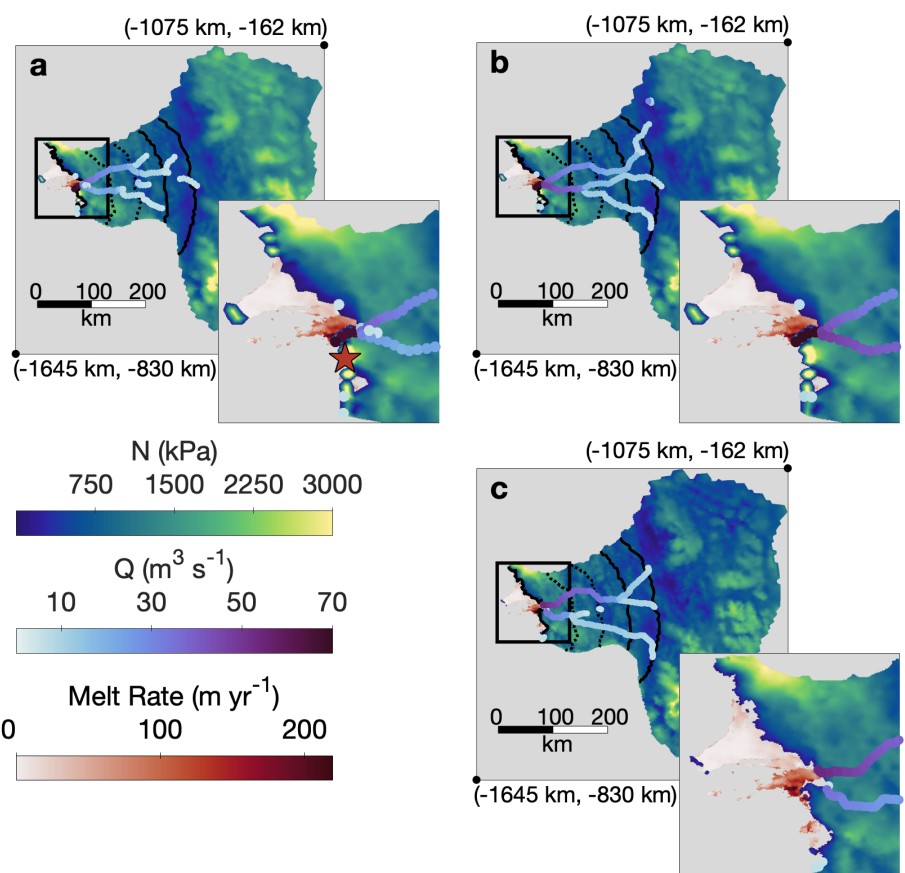

**Figure 5. a)** Average effective pressure and channel discharge across all data-compatible FSS runs. **b–c)** Effective pressure and channel discharge for **c)** the high-resolution model, and **b)** its low-resolution counterpart. The insets are enlarged views of the black boxes, and the star in **a** indicates the location of the secondary channel seen in one data-compatible FSS run. Sub-ice-shelf melt rates from Adusumilli et al. (2020) are plotted in all frames. For clarity, only channels with $Q > 5 \, \text{m}^3 \, \text{s}^{-1}$ are pictured in each frame. Again, transects spaced every 50 km from the terminus (used for determination of flux steady-state and in Figure 6) are shown as black lines, with the dotted lines spanning the transition zone of Schroeder et al. (2013). The locations of map corners are given in Standard Antarctic Polar Stereographic coordinates.

295    km transect is the first at which channelized drainage slightly outweighs distributed drainage, with discharges of $55 \pm 21$ and $47 \pm 20 \, \text{m}^3 \, \text{s}^{-1}$, respectively (Figure 6). Consistent with Joughin et al. (2009), basal friction melting is the primary contributor of melt in our model, and the 50–100 km transition to channelized flow coincides with a substantial increase in basal friction melt rate (Figures 1, 5, and 6) .

     Channelized discharge grows rapidly within 50 km of the terminus. By the point at which water reaches the grounding line,

channelized drainage accounts for $127 \pm 24 \, \text{m}^3 \, \text{s}^{-1}$ of runoff into the ocean, whereas only $25 \pm 21 \, \text{m}^3 \, \text{s}^{-1}$ is expelled through the distributed system (Figure 6). In all data-compatible FSS runs, the majority of channel discharge at the grounding line occurred through one primary channel with a discharge of $80 \pm 24 \, \text{m}^3 \, \text{s}^{-1}$ near the center of the grounding line ($-1.5369 \times 10^6$

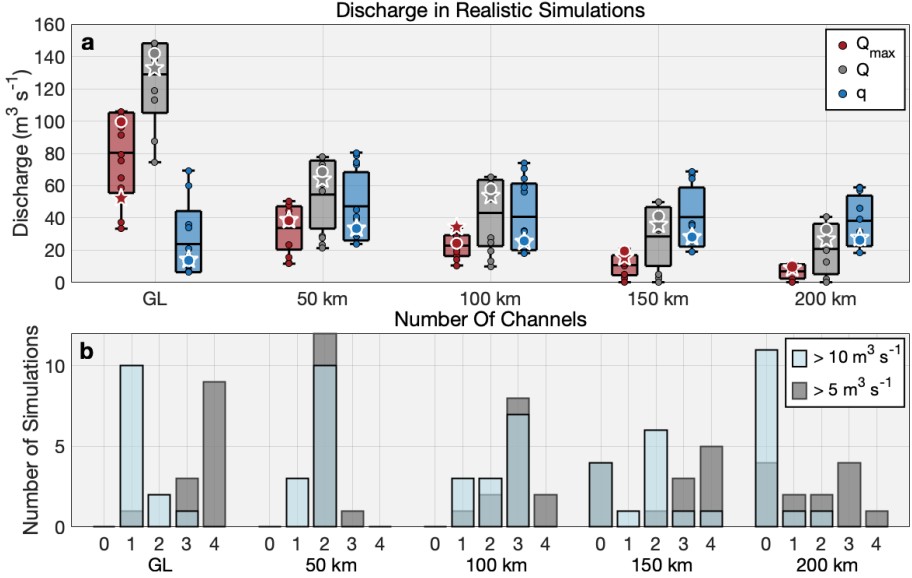

**Figure 6. a)** Total distributed (blue) and channel discharge (gray), as well as the discharge of the largest channel (red) across each transect (see Figure 5) for all data-compatible FSS runs (circles). Boxplots indicate the maximum, minimum, mean, and standard deviations. The stars indicate the high-resolution model, and the white-edged circles designate its low-resolution counterpart. **b)** The number of channels with $Q > 5 \text{ m}^3 \text{ s}^{-1}$ (gray) and $Q > 10 \text{ m}^3 \text{ s}^{-1}$ (blue) at each transect for all data-compatible FSS runs.

m, $-4.7298 \times 10^5$ m; Standard Antarctic Polar Stereographic). This location corresponds to the region of high basal melting observed at the Thwaites Ice Shelf in Adusumilli et al. (2020) (Figure 5). In one simulation, a secondary channel intersects the grounding line with a discharge of $38 \text{ m}^3 \text{ s}^{-1}$ at $(-1.5310 \times 10^6 \text{ m}, -4.8585 \times 10^5 \text{ m})$ where we lack basal melt data (Figure 5a). Other channelized discharge across the grounding line occurs through very small channels ($\lesssim 10 \text{ m}^3 \text{ s}^{-1}$ scattered along the marine boundary.

### 3.2 Grid Resolution Sensitivity Analysis

One data-compatible FSS simulation ($k_Q = 0.05 \text{ m}^{7/4} \text{ kg}^{-1/2}$, $k_q = 4 \times 10^{-4} \text{ m}^{7/4} \text{ kg}^{-1/2}$, $c_s$ 0.5 m$^{-1}$, $W_r = 0.1$ m) was rerun to flux steady-state with the high-resolution domain. The high-resolution run matched observed specularity content, and produced effective pressures and water fluxes that closely resembled its low-resolution counterpart. High-resolution channels followed very similar pathways as those in the low resolution model (Figure 5b–c), and distributed and channelized discharges at each transect were approximately equal to those at low-resolution (Figure 6a). The main exception occurred at the grounding line, where the two main channels reached the ocean independently in the high-resolution model, but merge just above the grounding line with lower resolution (Figure 5b–c). This explains the almost twofold discrepancy of maximum channel discharge at the grounding line between the two resolutions (Figure 6a). Additionally, the high-resolution run had lower effective

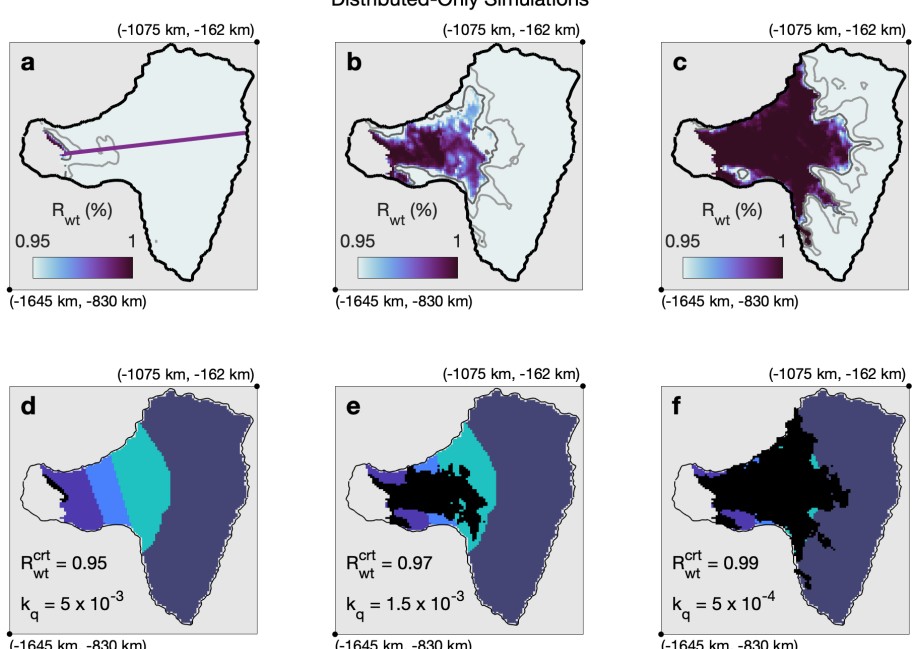

**Figure 7.** Three typical **a–c)** $R_{wt}$ configurations and **d-f)** coinciding binary masks (black) for distributed-only runs. Masks depict regions where $R_{wt}$ is above its threshold value, and thus the distributed system is at or above its capacity. $R_{wt}$ in distributed-only runs generally resembled one of these three patterns. Light and dark gray lines in **a-c** are the 50% and 90% $R_{wt}$ contours, respectively. Color-coding in **d-f** corresponds to the same zones as Figure 3. Purple line in **a** is the center-line transect used in Figure 8. $k_q$ used in each run, along with the $R_{wt}^{crt}$ used to create the coinciding binary mask, are provided in **d-f**. All three runs had bed roughness parameters $W_r = 0.1$ m and $c_s = 0.5$ m$^{-1}$.

pressures near the upper domain boundary, although effective pressures within 300 km of the terminus are in strong agreement with the low-resolution model (Figures 5, 8a).

### 3.3 Distributed-only Model Configuration

Average water pressures in our 25 distributed-only simulations ranged from 74-98% flotation, and all met our flux steady-state criteria. However, no distributed-only run had a $R_{wt}$ field that matched observed specularity content. In particular, the greatest mismatch occurred between $0 - 50$ km and $100 - 150$ km of the grounding line, where $R_{wt}$ was consistently over $R_{wt}^{crt}$, but where observed specularity content was low (Figure 7). In other cases where the average flotation percentage was below 90%, water thicknesses were too low to produce any regions of $R_{wt} \geq R_{wt}^{crt}$. Furthermore, distributed-only simulations had

unrealistically low effective pressures within 150 km of the terminus. Of the runs with an average water pressure over 90% flotation, many were at or near flotation within 200 km of the terminus (Figure 8a). Within 50 km of the terminus, the average effective pressure across these distributed-only runs was one-third that of data-compatible channel-enabled scenarios.

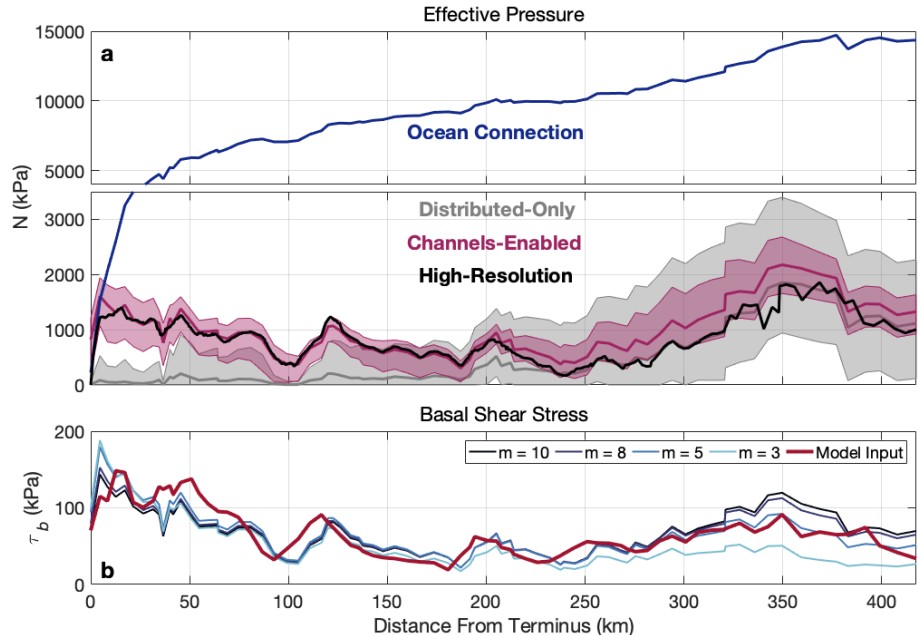

**Figure 8. a)** The range and mean (solid line) of effective pressures along the center-line transect in Figure 7a for all data-compatible FSS, channel-enabled runs (magenta) and all distributed-only runs above 90% flotation (gray). The black line depicts transect effective pressures from the high-resolution run. Shown in blue is the calculated effective pressure if assuming a perfect hydrostatic connection with the ocean. Note the different y-axis scale in the upper panel. **b)** Basal shear stress used as input in our model (red) plotted with reconstructed basal shear stress using a Budd-style friction law (blue). Blue hues represent different exponents used in the friction law. All lines follow the same center-line transect as in **a**.

An example comparison of catchment-scale features identified with binary masks (black) of observed specularity content and modeled $R_{wt}$. **a)** radar specularity content (Schroeder et al., 2013) and **c)** $R_{wt}$ for a data-compatible flux steady-state model run, together with their coinciding binary masks, **b)** ($S^{crt} = 0.19$) and **d)** ($R_{wt}^{crt} = 0.98$), respectively. The pink dashed line in **a** marks the transition between highly-specular, distributed drainage and channel-dominated drainage, as hypothesized in Schroeder et al. (2013). The four zones used for comparison between specularity content and $R_{wt}$ are color-coded in **b** and **d**. Light and dark gray lines in **c** are the 50% and 90% $R_{wt}$ contours, respectively. The percent match between masks within each zone and the overall correlation are given in **d**. The locations of map corners are given in Standard Antarctic Polar Stereographic coordinates.

## 4 Discussion

### 4.1 A Reconciled Framework for Channelization Beneath Thwaites Glacier

The key result of our study is the likely existence of stable subglacial channels beneath Thwaites Glacier. In our model, channels typically extended over 100–200 km inland, and had grounding line discharges of $80 \pm 24$ m$^3$ s$^{-1}$, much larger than the maximum discharges of 1–5 m$^3$ s$^{-1}$ and $< 25$ m$^3$ s$^{-1}$ modeled at Getz (Wei et al., 2020) and Totten (Dow et al., 2020) glaciers, respectively. No distributed-only experiments matched observed specularity content, and all had unrealistically high water pressures within 100 km of the terminus. This strongly argues that channelized drainage is necessary to explain observed radar specularity content.

Certain geometric and hydrologic conditions at Thwaites Glacier are unfavorable to the development of subglacial channels, and thus the extent of channelization in our model is somewhat surprising. In theory, subglacial channels should develop when the distributed system reaches a critical discharge that is inversely proportional to the hydropotential gradient (Schoof, 2010; Hewitt, 2011). In Greenland, it is believed that glaciers are unable to reach this critical threshold farther inland where gentle surface slopes weaken the hydropotential gradient and thick ice may expedite creep closure (Chandler et al., 2013; Meierbachtol et al., 2013; Dow et al., 2014). Similar logic could also apply to the thicker and broader Antarctic ice sheets, especially given their insignificant surface melt input. Yet, our model consistently depicts subglacial channels extending 100–200 km inland in all parameter choices. These channels could be explained by the large catchment size (189000 km$^2$) of Thwaites Glacier (Joughin et al., 2009), its funnel-like geometry, and high basal melt rates of 3.5 km$^3$ yr$^{-1}$ (Joughin et al., 2009), which together accumulate enough water to exceed the critical discharge threshold within 100–200 km from the grounding line. At first, the critical discharge may only be met locally (e.g., Hewitt, 2011) through the accrual of water in topographic depressions, which the subglacial channels tend to follow. High basal friction melt rates of $100 - 1000$ mm yr$^{-1}$ in the terminal 100 km, as calculated for our model input and by Joughin et al. (2009), are then likely responsible for the increased channelization near the grounding line.

Previous work has offered contrasting hypotheses on the persistence of subglacial channels beneath Thwaites Glacier. Originally, Schroeder et al. (2013) argued radar scattering from widespread concave channels produced the near-terminus, non-specular region they observed. However, an extensive channelized system may not allow for the isolation of subglacial lakes, and the discovery of subglacial lakes beneath Thwaites Glacier suggested channels may be ephemeral, forming only during subglacial lake drainage events (Smith et al., 2017). Based on our model, we here present a refinement of the hypothesis of Schroeder et al. (2013) that leaves room for the development of the subglacial lakes observed by Smith et al. (2017).

In agreement with Schroeder et al. (2013), we interpret the overlapping regions of observed high specularity content and high $R_{wt}$ between $100 - 250$ km from the terminus to unequivocally indicate the pooling of broad, flat water bodies in a distributed system near or at its capacity. This distributed-dominated system then transitions to a channel-dominated system between $50 - 100$ km from the terminus. Schroeder et al. (2013) hypothesized this transition to channelized flow occurs through the development of many channels spread across the glacier width, which scatter radar energy and lower specularity; however, our modeling instead suggests that the near-terminus, non-specular zone of Schroeder et al. (2013) depicts a below-capacity

distributed system, whose water has been partially evacuated by a small number of large, stable channels. Such a configuration would produce non-specular radar returns due to a rough surface of discontinuous water cavities at a variety of orientations (Figure 2).

In such a sparsely channelized system, it is expected that isolated areas of the bed exist in which subglacial lakes may form. Disconnected portions of the drainage network are common beneath alpine and Greenland glaciers, particularly in the summer

when channels draw water from the surrounding distributed system, leading to the isolation of poorly connected basal cavities (Murray and Clarke, 1995; Gordon et al., 1998; Andrews et al., 2014; Hoffman et al., 2016; Chu et al., 2016; Rada and Schoof, 2018). Disconnected areas may exist year-round, or may reconnect following a reconfiguration of the channelized system or the collapse of channels in the winter (Hoffman et al., 2016; Rada and Schoof, 2018). However, substantial subannual reshaping of the drainage system should not occur in the absence of a seasonal melt cycle, like at Thwaites Glacier, and thus parts of the

bed may remain disconnected for extended periods of time. This would allow disconnected water to gradually pool into lakes that drain when they periodically exceed their hydropotential seal (Fowler, 1999). Such drainage events could act as similar catalysts for drainage network reconfigurations as the seasonal melt cycles of alpine and Greenland glaciers. Our model lacks the complete physics to properly simulate the filling and draining of subglacial lakes (e.g., Carter et al., 2017); however, it is evident that persistent and extensive subglacial channels can exist concurrently with subglacial lakes beneath Thwaites Glacier,

and further work is needed to understand the interaction between the two drainage features.

## 4.2 Implications of Channelization on Thwaites Glacier Dynamics

### 4.2.1 Channelization and Submarine Melting at the Grounding Line

The rapid and potentially unstable retreat of Thwaites Glacier is likely driven by enhanced sub-ice-shelf melting (Rignot et al., 2014; Joughin et al., 2014; Seroussi et al., 2017; Yu et al., 2018; Milillo et al., 2019; Hoffman et al., 2019), resulting in part

from intruding warm Circumpolar Deep Water (CDW) flowing along bathymetric troughs to the grounding line (Nakayama et al., 2019; Milillo et al., 2019; Hogan et al., 2020). The most rapid retreat (12–18 km between 1992–2011) was recorded at the glacier's central, fast-flowing core (Rignot et al., 2014), where the retreat has continued at a rate of 0.6 km yr$^{-1}$ until at least 2017 (Milillo et al., 2019). Ice shelf submarine melt rates exceed 200 m yr$^{-1}$ at the fast-flowing core, coincident with the recent formation of a prominent sub-shelf cavity (Adusumilli et al., 2020; Bevan et al., 2021).

In all but one of our low-resolution data-compatible FSS runs, both main channels converge near the grounding line directly above the subshelf cavity described in Bevan et al. (2021) (Figure 5a–b). In our high-resolution model, one channel intersects the grounding line at this location, while the second reaches the ocean 16 km to the east, also in the region of high subshelf melting (Figure 5c). Subglacial discharge plumes, formed from channelized subglacial water entering the ocean, amplify local submarine melting through turbulent heating and the entrainment of deep and often warm water, such as CDW, along the

terminus and ice shelf (Jenkins, 2011; Slater et al., 2015; Asay-Davis et al., 2017). While it would be an over-interpretation of our model to regard the exact locations of subglacial channels as reality, the ubiquitous conjunction of large channels (33–106 m$^3$ s$^{-1}$) with high subshelf melt rates at the grounding line in all data-compatible scenarios strongly suggests channelized

subglacial discharge augments submarine melting in this region. Recent ocean modeling of the Pine Island Ice Shelf cavity supports this assertion, and indicates subglacial discharge localized at the grounding line, and of similar magnitude to what occurs in our model, can explain the local ice shelf melt rates of $\sim 200$ m yr$^{-1}$ observed at Pine Island Glacier (Nakayama et al., 2021). Similar results have also been reported for the nearby Getz Ice Shelf, where subglacial discharge accelerates subshelf submarine melting by entraining and displacing CDW along the base of the ice shelf (Wei et al., 2020).

Additionally, CDW reaches the Thwaites Glacier grounding line through a series of bathymetric troughs and sills that moderate its flow (Nakayama et al., 2019; Hogan et al., 2020), and it is possible the entrainment of ambient water into subglacial discharge plumes may further enhance CDW flushing of the Thwaites subshelf cavity, similar to the subglacial plume-driven renewal of Greenland fjords (Gladish et al., 2015; Carroll et al., 2017; Zhao et al., 2021). However, plume-driven buoyancy forcing may only have a minimal effect on cavity circulation beneath the Pine Island Ice Shelf (Nakayama et al., 2021), and thus it could be assumed that the comparable grounding line fluxes given by our model are still too weak to significantly enhance CDW advection to Thwaites Glacier.

### 4.2.2 Implications of Channelization for Effective Pressure and Basal Sliding

Despite contributing to high ice-shelf basal melt rates and potential loss of ice-shelf buttressing, our model suggests subglacial channels may have a stabilizing effect on basal drag near the grounding line. Effective pressures are $3\times$ higher within 50 km of the grounding line in channel-enabled runs than in distributed-only runs (Figure 8d). This region of high effective pressure coincides with a distributed system that is operating below its capacity (Figure 3), something not reproducible in distributed-only simulations (Figure 8a–c). Only 1–3 principal channels exist within the terminal 100 km; nevertheless, comparison with distributed-only experiments indicates that a small number of channels are still able to efficiently evacuate water from the entire region, due to their lower hydropotential compared to the surrounding area. Higher effective pressure in the terminal 100 km implies higher basal friction, which has been shown to be a leading control on the retreat and mass loss of Thwaites Glacier (Yu et al., 2018) and surface velocities at the neighboring Pine Island Glacier (Gillet-Chaulet et al., 2016; Joughin et al., 2019). High basal shear stress associated with competent bedrock is already thought to exist within 80 km of the grounding line (Joughin et al., 2009), and may work in tandem with channelized subglacial drainage to help buttress against further retreat.

Effective pressures decrease substantially further inland where channelization is minimal. In the upper highly specular area, average effective pressures in data-compatible runs range between 200–600 kPa, almost an order of magnitude less than the near-terminus region (Figure 8). Effective pressures in highly specular areas are similar to the -30–150 kPa effective pressures observed at Ice Stream B (Engelhardt and Kamb, 1997), which to our knowledge, remain the only direct observations of effective pressures in West Antarctica.

Smith et al. (2017) noted that the small ($< 10\%$) increase in ice velocity observed after subglacial lake drainage events may indicate an insensitivity of Thwaites Glacier dynamics to its subglacial hydrology. However, the linked subglacial lake drainage event measured by Smith et al. (2017) beneath Thwaites Glacier in 2013–2014 had an average discharge of $160 - 240$ m$^3$ s$^{-1}$ over 6 months; only 3–5 times greater than modeled channel discharge 50 km from the terminus, and 1–2 times greater than the largest modeled channels at the grounding line. Any pre-existing channels of similar size to those in our model could,

therefore, help accommodate the additional flux from lake drainage events, which may explain the relatively minor increase in ice velocity they observed. Thus, this lake drainage event could also be interpreted as evidence of channelized drainage stabilizing glacier dynamics, as is indicated by our model. As Thwaites Glacier continues to thin and retreat, we expect the subsequent changes in glacier geometry and meltwater input to continually reshape its subglacial drainage network. Our results suggest this will alter ice dynamics, and should be taken into account when considering the uncertainty in model projections.

Ice dynamics models have recently started implementing effective pressure-dependent sliding laws supported by current theory. However, a challenging problem is how to best parameterize effective pressure in order to solve for basal shear stress. A common approach is to approximate effective pressure by assuming a perfect hydrostatic connection with the ocean (e.g., Leguy et al., 2014; Asay-Davis et al., 2016; Yu et al., 2018; Nias et al., 2018; Cornford et al., 2020, and others), shown for our model domain in Figure 8a. Effective pressure using an ocean connection assumption is in fair agreement with our channel-enabled runs within 5 km of the grounding line, but is up to an order of magnitude too high further inland, indicating a parameterization based on an open ocean connection may only be realistic near the terminus. This suggests a regularized-Coulomb friction law (e.g., Joughin et al., 2019) may be appropriate for Thwaites Glacier, as it only accounts for effective pressure where effective pressure is low and basal sliding speeds are high, such as near the grounding line (Schoof, 2005). However, our channel-enabled model indicates effective pressure actually decreases between 5–100 km from the grounding line, and maintains its proportionality to basal shear stress throughout the entire domain (Figure 8b). This implies basal shear stress stays in the Coulomb regime even within the glacier interior, and thus a yield stress or semi-plastic Budd-type law may work equally well for Thwaites Glacier, as has previously been successful at Pine Island Glacier in reproducing observed surface velocities (Gillet-Chaulet et al., 2016).

To test this hypothesis we attempt to reconstruct our input basal shear stress using a Budd-style friction law of the form: $\tau_b = CN\boldsymbol{u}_b^{1/m}$, where $\boldsymbol{u}_b$ is a model input, $N$ is solved for by the hydrology model, and $C$ is a tunable basal slipperiness coefficient. Here, $m$ is the bed-dependent stress exponent that is likely between 5–10 for Pine Island Glacier (Gillet-Chaulet et al., 2016; Nias et al., 2018; Joughin et al., 2019), which is assumed to have similar basal properties to Thwaites Glacier. Figure 8b illustrates the results using four plausible values of $m$ and accompanying $C$ values that minimize the root mean square error with the model input. All four versions effectively recover the input basal shear stress, with the best agreement using $m = 5$ or $m = 8$, which is consistent with previous work (Gillet-Chaulet et al., 2016; Nias et al., 2018; Joughin et al., 2019). Therefore, we assert that a Budd-style friction law is appropriate for Thwaites Glacier, assuming accurate knowledge of the effective pressure field. Based on these results we caution against the continued usage of the hydrostatic ocean connection parameterization for effective pressures beyond the marginal 5 km for Thwaites Glacier, which may produce unrealistically slow sliding velocities.

### 4.3 Model Considerations

Our results highlight the need for validation of subglacial hydrology models across the entirety of a glacier. We found a wide range of parameter values resulted in steady-state configurations, and most had some degree of channelization coincident with the location of observed anomalously high sub-ice-shelf melting. However, many simulations had water pressures and

discharges that were either too low or too high to be realistic, and without comparison with radar specularity content, it would have been easy to arbitrarily choose the wrong parameters and base our conclusions on an unrealistic model. Borehole validation has been previously attempted for a small alpine glacier (Rada and Schoof, 2018), but the scale of Antarctic and Greenland glaciers makes this unattainable for ice sheets. We therefore suggest that ice-penetrating radar, such as used in this paper and in Dow et al. (2020), or other broad-scale proxies for basal water, is the best approach for validation of ice sheet subglacial hydrology models. While our comparison between $R_{wt}$ and specularity content is somewhat *ad hoc*, it selected for a coherent grouping of parameters, water pressures, and channel velocities within the expected realistic range, which gives us confidence in its effectiveness. Comparison criteria may need customization to be applicable at other glaciers, but the overall methodology presented in this paper should be beneficial in many settings. Bed conditions differ within and between glacier basins, and we stress our parameter choices should not be extrapolated to other glaciers without validation.

Many assumptions built into subglacial hydrology models remain unsupported, and it is uncertain how such assumptions may influence our results. We therefore deem it necessary to consider the primary underlying simplifications that may impact this paper. Our choice to ignore pressure-dependent melting/freezing in Equation 5 neglects the effects of supercooling, which would lead to the abatement of R-channels and the expansion of the distributed system as water flows out of a prominent overdeepening. Supercooling has been shown to decrease channelization in other subglacial hydrology models (de Fleurian et al., 2018). However, the overdeepening within $100\,\mathrm{km}$ from the grounding line (Figure 1a), in which channelization becomes pronounced, is far from meeting the supercooling threshold of Werder (2016). Furthermore, the upward bed slope in the terminal $100\,\mathrm{km}$ is only $60\%$ of the downward surface slope, and should therefore allow for sufficient dissipative heating to continually grow channels (Alley et al., 1998). We therefore do not expect the neglect of $\Pi$ in Equation 5 to significantly affect our conclusions.

Uniform parameterizations of the distributed system do not account for realistic heterogeneity in bed geometry or lithology, both of which can locally influence distributed connectivity (Murray and Clarke, 1995; Gordon et al., 1998; Andrews et al., 2014; Hoffman et al., 2016; Rada and Schoof, 2018; Downs et al., 2018). The bed of Thwaites Glacier is thought to consist of alternating regions of bedrock and glacial till (Joughin et al., 2009; Muto et al., 2019a, b; Holschuh et al., 2020) that could potentially affect the connectivity of the distributed system, and thus conductivity and discharge. Currently, all subglacial hydrology models assume a consistent $k_q$ across their domains, although allowing $k_q$ to vary with bed lithology may account for spatial differences in connectivity and produce more realistic results (Hoffman et al., 2016).

Modeling (Joughin et al., 2009) and seismic data (Muto et al., 2019b, a) suggest bed elevation could serve as a reasonable proxy for bed lithology under Thwaites Glacier, where subglacial till (low conductivity) accumulates in depressions and exposed bedrock (high conductivity) primarily exists at topographic highs. Regions of high specularity content coincide with low-lying troughs, and it is therefore conceivable that imposing a high $k_q$ above these troughs, and low $k_q$ within them, could reproduce the observed specularity content without the need for channelization. However, our results suggest the minimum $k_q$ necessary to prevent channelization would still be high enough over a majority of the domain to drop water pressures below realistic levels. Lowering $k_q$ within troughs, but maintaining the same $k_q$ at higher elevations as used in our data-compatible FSS runs could help pool water into subglacial lakes in till-laden depressions (see Section 4.1), but it seems unlikely this would

divert enough water to preclude the overall growth of channels in the terminal 100 km. Furthermore, the location of modeled channelized flow at the grounding line presents a convincing explanation for the anomalously high sub-ice-shelf melt rates observed at the same position, something that would be lacking in a purely distributed system. We acknowledge the neglect of a spatially variable $k_q$ could create some uncertainty in our discharge results, but is likely minimal, and our $k_q$ parameter sweep may already account for this variability.

As described in Downs et al. (2018), the value of $k_q$ used in subglacial hydrology models is a proxy for the connectivity of orifices linking cavities in the bed. Models assume the orifices scale with cavity size; however, in their original conception, orifices behave like small R-channels that may enlarge with turbulent melting (Kamb, 1987; Fowler, 1987). Downs et al. (2018) used this argument to scale $k_q$ with meltwater input, which better captured seasonal water pressures. Although Thwaites Glacier lacks a seasonal meltwater cycle, we could use the same argument to justify use of a different distributed system flow law.

Darcy or Darcy-Weisbach flow laws are used almost ubiquitously in subglacial hydrology models (e.g., Schoof, 2010; Hewitt, 2011; Werder et al., 2013; Hewitt, 2013; Hoffman and Price, 2014; Downs et al., 2018; Hoffman et al., 2018; de Fleurian et al., 2018; Dow et al., 2020, and others), yet these laws are largely unvalidated in the subglacial environment. Distributed discharge with a Darcy-Weisbach turbulent flow law, as used in this paper, has a $\frac{5}{4}$ power dependency with water thickness. However, in other flow laws, such as Darcy porous media flow or Poiseuille laminar flow, the exponent may vary between 1 and 3 (e.g., Hewitt, 2011, 2013; Kyrke-Smith and Fowler, 2014; Kyrke-Smith et al., 2014). In practice, the use of a higher exponent could produce similar behavior to a melt-dependent $k_q$, and could account for a larger connectivity with increased meltwater, driven by the dissipative melting and opening of orifices. Although such a flow law would increase efficiency of the distributed system and potentially minimize channelization, we do not believe its use would dramatically change our results. Water thicknesses using a Darcy-Weisbach law are fairly uniform within 200 km of the grounding line (Figure 3c), which suggests an increased dependency of discharge on water thickness may make little difference in our model.

## 5 Conclusions

This paper leverages observations from a variety of sources to select for the subglacial hydrology model scenarios that are the most likely representations of reality. Our range of possible steady-state scenarios highlights the need for thorough parameter sweeps in subglacial hydrology models, which are then winnowed to the most realistic grouping of simulations based on extensive observations. We emphasize validation of subglacial hydrology models within the glacier interior, and not just at its terminus, is necessary to properly constrain realistic drainage behavior. Furthermore, our work demonstrates subglacial hydrology models still produce a range of results that are compatible with data, and thus model results should be reported as a suite of possible scenarios, instead of one feasible configuration.

Our work presents an updated conceptual model for the subglacial drainage system beneath Thwaites Glacier. Our model indicates a few stable channels exist within 200 km of the grounding line, and coalesce into 1–2 large stable channels within the terminal 50–100 km. These channels intersect the ice-ocean boundary directly at the location of highest sub-ice-shelf melt rates, suggesting they play an important role in frontal ablation and grounding line retreat. However, in the interior of the

glacier, subglacial channels efficiently evacuate water from a broad portion of the bed, thereby increasing basal friction within 100 km of the grounding line and potentially buttressing against further retreat. At this point, it remains unclear how common such drainage systems are in Antarctica, or what impact subglacial channels have on sub-ice-shelf cavity circulation and ice dynamics. We expect the subglacial drainage network to continually reconfigure with future changes in meltwater production and glacier geometry, which will subsequently lead to spatially and temporally evolving basal shear stress and frontal ablation rates. Further work with a fully coupled ice dynamics-subglacial hydrology model will be necessary to determine the exact influence of subglacial channels on future retreat and mass loss.

*Code and data availability.*  Model output, radar data, processing files, and the MALI model code used to perform the simulations described can be obtained online at https://doi.org/10.5281/zenodo.5593376.

### Appendix A:  Parameter Sweep, Sensitivity Analysis, and Steady-State Criteria

A full sweep of realistic conductivity parameter space was conducted for each set of bed roughness parameters; however, our method for determining bed roughness parameters was closer to that of a sensitivity analysis (varying one parameter at a time). This choice was made because real physical constraints exist for conductivity parameters, while bed roughness parameters are theoretical quantities approximating general bed characteristics that only have indirect physical corollaries. A sensitivity analysis is thus more suitable for bed roughness parameters and allowed us to ease the complexity of sampling a four-dimensional parameter space. Results for each conductivity parameter sweep (in addition to Figure 4) are depicted in Figures A1 – A5.

Establishing steady-state criteria inherently involves defining a cutoff threshold for acceptable noise remaining in the model. For our pressure steady-state runs, effective pressure at each cell is allowed to fluctuate $0.5\%$ of its value on average. This equates to an allowable fluctuation of roughly 1 kPa where effective pressure is lowest ($\sim 200$ kPa) and 10 kPa where effective pressure is highest ($\sim 2000$ kPa). For flux steady-state runs, meltwater production above each transect must equal the total discharge across the transect within $0.5\%$. Total melt production above the grounding line is roughly 155 $\mathrm{m^3 s^{-1}}$, so our flux steady-state criteria require that we know the total grounding line discharge within 0.8 $\mathrm{m^3 s^{-1}}$, which is orders of magnitude less than the uncertainty between data-compatible FSS runs.

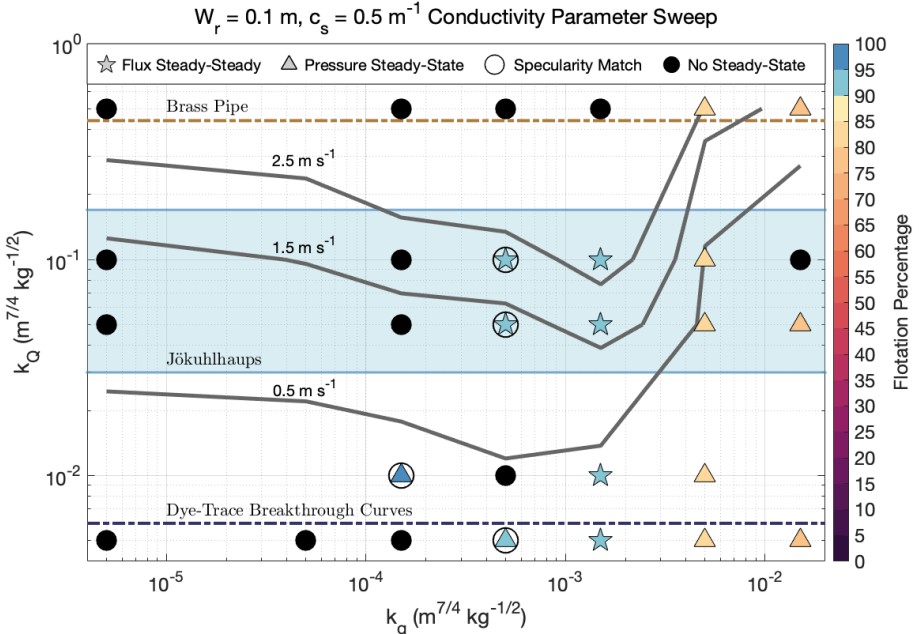

**Figure A1.** Same as Figure 4, but for bed roughness parameters $W_r = 0.1$ m and $c_s = 0.5$ m$^{-1}$. Stars represent runs that reached flux (and pressure) steady-state, triangles symbolize pressure steady-state simulations, and filled black circles depict runs that did not reach either steady-state criterion. Symbols for steady-state runs are color-coded by the average flotation percentage of grounded ice. Circles around stars or triangles indicate runs that matched observed specularity content, and are considered data-compatible. Gray lines are $95^{th}$ percentile channel velocity contours for channels with $Q > 5$ m$^3$ s$^{-1}$. $k_Q$ limits determined from a brass pipe and dye-trace breakthrough curves are plotted as brown and dark-blue dashed lines, respectively, and the blue shaded area represents the typical observed Jökulhlaup $k_Q$ range.

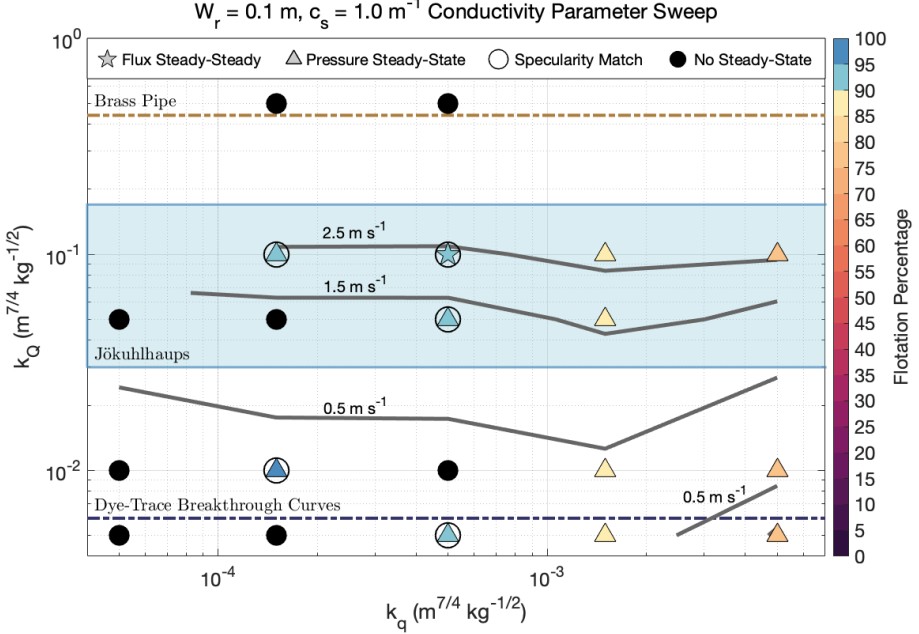

**Figure A2.** Same as Figure 4, but for bed roughness parameters $W_r = 0.1$ m and $c_s = 1.0$ m$^{-1}$.

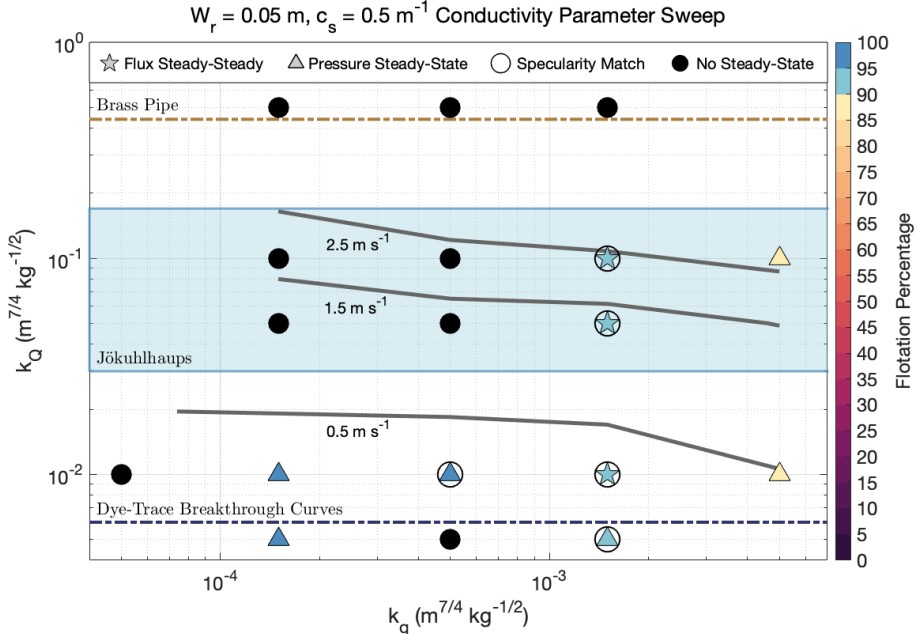

**Figure A3.** Same as Figure 4, but for bed roughness parameters $W_r = 0.05$ m and $c_s = 0.5$ m$^{-1}$.

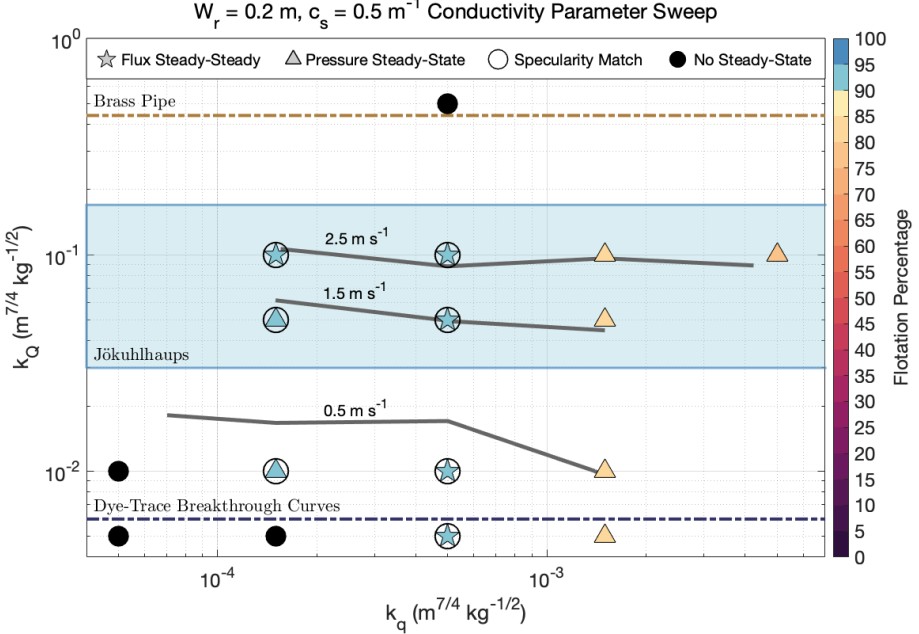

**Figure A4.** Same as Figure 4, but for bed roughness parameters $W_r = 0.2$ m and $c_s = 0.5$ m$^{-1}$.

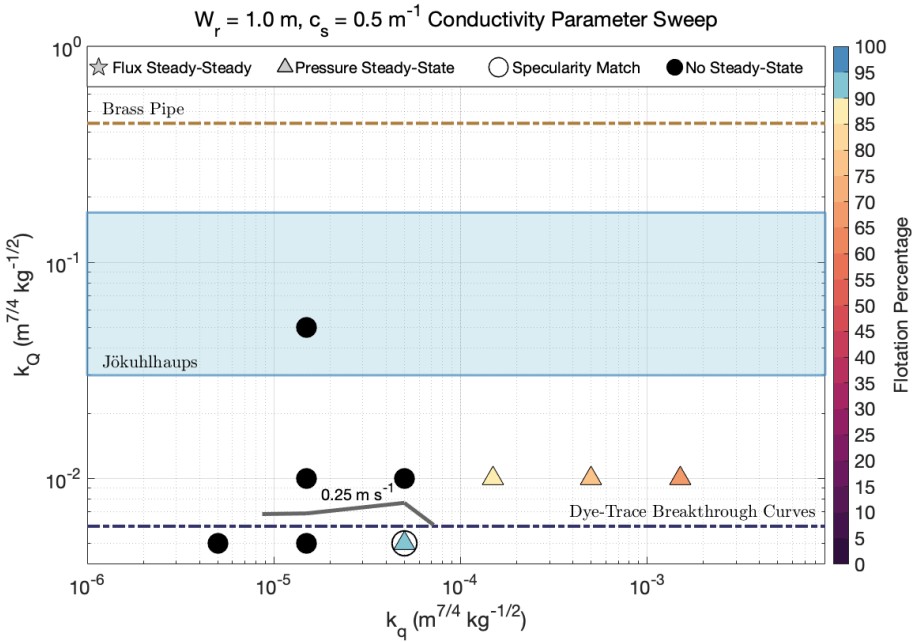

**Figure A5.** Same as Figure 4, but for bed roughness parameters $W_r = 1.0$ m and $c_s = 0.5$ m$^{-1}$.

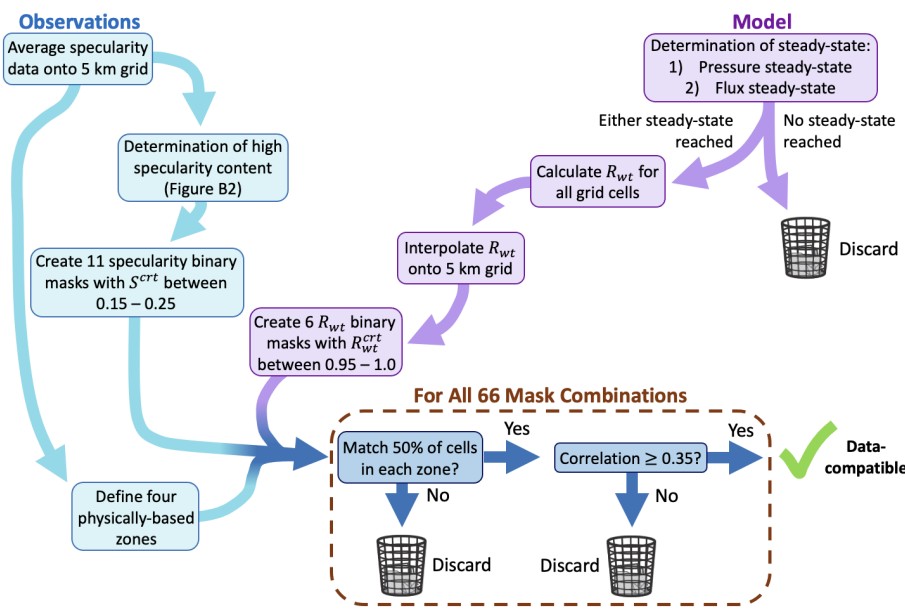

**Figure B1.** Flow chart illustrating the set-by-step process for determining which model runs were compatible with observed specularity content.

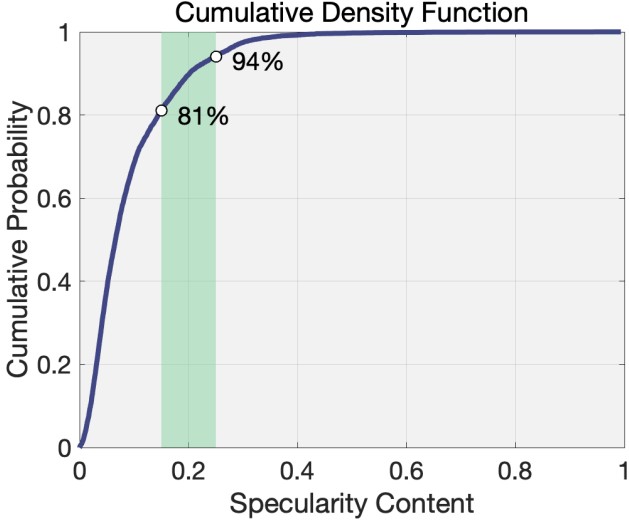

**Figure B2.** Cumulative density function of observed specularity data from Schroeder et al. (2013). The green band highlights the range of specularity values used to create our 11 specularity masks, which are in the 81[st] to 94[st] percentile of our dataset.

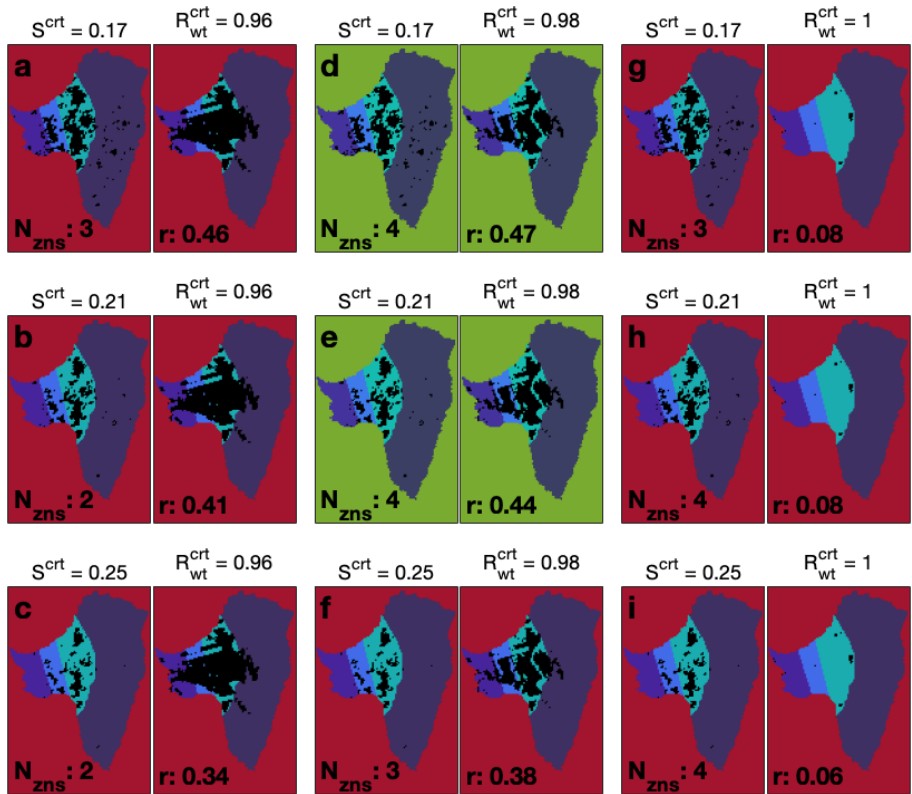

**Figure B3.** Select $S^{crt}$ (observed) and $R_{wt}^{crt}$ (modeled) mask combinations from the $W_r = 0.1$ m, $c_s = 0.5$ m$^{-1}$, $k_q = 5 \times 10^{-2}$ m$^{7/4}$ kg$^{-1/2}$, $k_Q = 5 \times 10^{-4}$ m$^{7/4}$ kg$^{-1/2}$ model run, plotted over the four zones used for the first comparison criterion (also shown in Figures 3, 7). $N_{zns}$ indicates the number of zones that meet Criterion 1, and $r$ is the overall correlation between mask pairs. Background color indicates successful (green) and unsuccessful (red) matches. Values of $S^{crt}$ and $R_{wt}^{crt}$ used to make each mask are displayed above each plot.

## Appendix B: Comparison Criteria Between Modeled and Observed Specularity Content

Matching specularity and $R_{wt}$ masks is a comparison between two spatial point patterns, which can be challenging as it requires a global statistic that can recognize local patterns of point clusters. Other comparisons of spatial point patterns have relied on segmenting the domain into areal units and determining an overall similarity statistic across all units (e.g., Andresen, 2009, 2016). The method developed in the current paper shares the concept of areal units by defining four physically-based zones within which we assess similarity between the two masks. These zones are intentionally chosen to loosely encompass regions of specularity or non-specularity, which allows for some spatial variability between masks and decreases the sensitivity to the zonal boundaries. We then require the two specularity and $R_{wt}$ masks to match at 50% or more of grid points within each zone. As low specularity can occur for a variety of reasons, segmenting the domain into specularity-based zones does not predetermine a specific drainage style, but preserves the specular pattern of interest and allows us to test hypotheses concerning its formation.

While the first criterion does well by itself in selecting positive matches, it also selects many false positives. This occurs when the $R_{wt}$ mask is almost entirely non-specular and over 50% of the cells in each zone is non-specular in the observed specularity mask (Figure B3 h–i). It was therefore necessary to include a second criterion that can remove these false positives, which we do by requiring an overall correlation coefficient of $r \geq 0.35$. Correlations are calculated with:

$$r = \frac{\sum_m \sum_n (S_{mn} - \bar{S})(R_{mn} - \bar{R})}{\sqrt{(\sum_m \sum_n (S_{mn} - \bar{S})^2)(\sum_m \sum_n (R_{mn} - \bar{R})^2)}} \tag{B1}$$

where $S$ and $R$ are the specularity and $R_{wt}$ masks, respectively. Again, correlation by itself does a fair job at identifying positive matches, but it also identifies false positives when the $R_{wt}$ mask is overly specular (Figure B3 a–b). As the two criteria fail for opposing reasons, they can check and balance each other if the thresholds are tuned appropriately (Figure B3 d–e). We acknowledge this comparison method is sensitive to multiple choices of thresholds, so we attempt to make our criteria for selecting data-compatible runs as generous and inclusive as possible while still removing runs that clearly do a poor job at resembling observations. We empirically determined that requiring $\geq 50\%$ of cells in each zone to agree and $r \geq 0.35$ works well at identifying positive matches and is sufficiently general to allow a reasonable variety of $R_{wt}$ masks to pass this filtering process.

*Author contributions.* AH and MH conceived of the study and designed the simulation plan. AH conducted the model simulations, developed and carried out the analysis, wrote the majority of the manuscript, and created the figures. MH also assisted with analysis and writing of the paper, designed and implemented the MALI subglacial hydrology model, and created the Thwaites model domain. SP established funding for the research, provided experience in ice sheet modeling, and gave extensive guidance throughout the research process. DS contributed his expertise in radar specularity analysis and interpretation, which was critical for comparison with model results. All authors contributed to editing the manuscript and discussing methodology.

*Competing interests.* The authors declare they have no competing interests in the publication of this article.

*Acknowledgements.* Support for this work was provided through the Scientific Discovery through Advanced Computing (SciDAC) program and the Energy Exascale Earth SystemModel (E3SM) project funded by the U.S. Department of Energy (DOE), Office of Science, Biological and Environmental Research, and Advanced Scientific Computing Research programs. This research used resources of the National Energy Research Scientific Computing Center, a DOE Office of Science user facility supported by the Office of Science of the U.S. Department of Energy under Contract DE-AC02-05CH11231, and resources provided by the Los Alamos National Laboratory Institutional Computing Program, which is supported by the U.S. Department of Energy National Nuclear Security Administration under Contract DE-AC52-06NA25396. This work was also partially supported by the US National Science Foundation Office of Polar Programs under Grant 1543012. We thank Dr. Trevor Hillebrand for processing the observational datasets used to generate the model initial condition and providing computing time on a Los Alamos National Laboratory Institutional Computing Program allocation. We thank Dr. Mauro Perego for his

contributing an optimized ice velocity and stress field for the model domain and for reviewing a version of the manuscript. We also thank our editor, Dr. Kang Yang, as well as Dr. Doug Brinkerhoff and our two anonymous reviewers for their insights for improving the manuscript.

This work developed out of a student project at the 2018 International Summer School in Glaciology in McCarthy, Alaska, organized by the University of Alaska Fairbanks.

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
