# Peer review of "Persistent, Extensive Channelized Drainage Modeled Beneath Thwaites Glacier, West Antarctica"

_The Cryosphere, 2021_

## Referee Comment (RC1)

Review of "Persistent, Extensive Channelized Drainage Modeled Beneath Thwaites Glacier, West Antarctica - Hager et al."

Summary of the paper:

The manuscript investigates the possibility of formation of channelized drainage under the Thwaites glacier using subglacial hydrology component of MPAS-Albany Land Ice model. Earlier, Schroeder et al. 2013 reported transition of water system from distributed to efficient drainage system in the interior of Thwaites glacier from specularity content data of bed echo. In this paper, the authors run several numbers of simulations by varying four parameters (sheet conductivity, channel conductivity parameter, Cavity spacing, and bed bump height) and compares the modelled results with the specularity content data of Schroeder et al. 2013 to assess the likelihood of channelization under Thwaites glacier.
The authors discuss their results with different geophysical properties of the Thwaites glaciers, some of which are important for further investigation. A counter-example analysis of channelization is also provided by running some simulations with channelization disabled which is a great addition. It is subject to discussion whether the findings are robust, but the presented results are very well complemented with counter examples, available data and adequate discussions, and that makes it worth getting published.

The paper is overall well written. The introduction nicely covers previous knowledge of subglacial hydrology in general and this region in particular. The results are very well-articulated. The authors complement their results with other observations such as ice shelf basal melt and present some interesting discussions about the findings of the paper, such as effective pressure consideration at the grounding line in sliding laws, etc., which is important for future research. Overall various geophysical aspects of Thwaites glacier are discussed under the findings of this paper, and all these make the content of the paper rich. However, the paper has some loose ends which requires some work. I mention them in the following:

Comments:

1. The method section, especially section 2.4 needs substantial work. This part is very much unclear. The physical basis of choosing the specularity content threshold is not clear to me. Similarly, it is not clear why 6 different Rwt threshold values are used instead of one Rwt (e.g. 0.95). At present it is not easy to understand how different simulations are done with different combinations of parameters. I would suggest to provide a table explaining the simulations. Additionally, it is not obvious how different parameter sweep leads to certain number of simulations.

2. Whereas the parameters choices are very well explained, the choice of thresholds lacks sufficient explanations. There are many thresholds used (flotation, flux steady state, Pressure steady state (all in sect. 2.3), Scrt, Rwt, correlation coefficient> 0.35 (sec. 2.4)) and results of this paper are highly dependent on the choices of these threshold. These choices remain very

subjective, and not enough supported analysis is provided for their choice. I would recommend to provide substantial logic for using those thresholds and have detail discussion around them. The authors need to present some more statistics to support these choices, and it can be included as appendix or as supplementary, if not as main article. That brings me to the question that how does your result are sensitive to the choice of threshold?

3. Comparison with specularity content seemed bit like cherry-picking. However, I do not deny the potential of specularity data in understanding subglacial hydrology. I just feel that these data can be used better/sophisticated way to infer status of subglacial hydrology. Results associated with specularity content are not very robust and presented in very sporadic manner. In my opinion, this is the major area of improvement for the manuscript. The authors should explore better way to have comparison with specularity content data. The choice of threshold of specularity data is not clear. There is no physical basis of it. Furthermore, the description needs to be improved substantially. At present, this part is not completely clear to me. I would recommend to add more detail description with figures for this section to enhance the readability.
Although I do not have any clear suggestions on specularity content data, but the authors should find a better way to compare the specularity data which I think does not require any additional model runs. The present way of representation and analysis is neither very convincing, nor easy to understand.

In addition, I have some major specific questions/comments:

1. Can you please elaborate why two different steady-states criteria are chosen?

2. Please provide justification of considering avg. water pressures of >90% flotation. Using 90% only is very subjective. It would be good to provide supporting result of choosing 90%. For example, show how your results will differ when using 80% or 95% flotation.

3. The effective pressure (N) in the interior seems bit high especially where specularity content is high which is supposed to represent distributed drainage! Can you have some discussions on your derived effective pressure value with effective pressure reported in other studies? You provide good discussion with the discharge from previous studies of Antarctica, but I would recommend to do the same for effective pressure.

4. In addition to above comment on sec. 2.4, I would recommend to add few figures showing different masks derived using thresholds of Scrt or Rwt. If not here, these figures must be provided in appendix or in supplementary.

5. The paper presents nice analysis with parameter sweep and completement their results with ice shelf basal melt rate. I think this part can get more focus as it is an interesting comparison (e.g., Wei et al., 2020).

Minor comments

Line 215: 'majority of the cells' - How many number or cells do you mean here?

L 229-230: Was zero instances of water pressure below 90% outside data-compatible run? I would suggest to include a table here too with these criteria. Otherwise, this line remains bit vague.

L 256- 257: " .. the 50–100 km transition to channelized flow coincides with a substantial increase in basal friction melt rate." - Can you please elaborate on that with some data?

Figures 2,6 - What does the black dots in b and d represents? Are these the locations of significant correlations?

Fig. 5: Does '> 5m2s-1' include >10 m3s-1? or it is >5 and <10?

L 275 -277: " .. pressures near the upper domain boundary, although effective pressures within 300 km of the terminus are in strong agreement with the low-resolution model." - It is not clear to me from the figure.

L561: The author list is incomplete.

---

## Referee Comment (RC2)

**Referee comment on 'Persistent, Extensive Channelized Drainage Modeled Beneath Thwaites Glacier, West Antarctica'**

Doug Brinkerhoff

January 2022

**1   Summary**

In this section, the authors make the case for significant channelization in the Thwaites catchment, which may have substantial implications for its evolution due to both the modification of sub-shelf circulation by melt plumes and also a relatively higher upstream basal shear stress due to efficient evacuation of water (and thus a reduction in water pressure). This argument primarily relies upon the comparison of a numerical model of drainage development with putative radar-based observations of a proxy for drainage type. A moderately large ensemble of model configurations are tested, leading to several configurations that are consistent with observations, all of which involve the development of substantial channelization.

I find this paper to be well written and compelling. I particularly appreciate the ensemble approach to the modelling, which greatly improves the robustness of the results. I do not have any major comments; however I do have a handful of minor comments, clarifications, and technical corrections that I would like to see addressed.

**2   Line-by-line comments**

**L92** I suggest mentioning that the implications of neglecting the pressure change term are discussed in the discussion.

**Eq. 7** Where are the channels? Are they defined on the edges of the Voronoi mesh, or between center points, or something else entirely?

**L101** I suggest language like 'time derivatives are discretized using an explicit forward Euler method ...'.

**L104** What are these 'transients' mentioned here? Numerical instabilities? Usually this word refers to non-physical changes induced by initial conditions that are far out of balance with the governing equations, in which case, temporal averaging wouldn't seem to do much.

**L117** 'surfface' → 'surface'

**L123** I don't understand the notion of varying hydropotential values: isn't sea level just sea level? Depth wouldn't have anything to do with it.

**L123** Why suppress backflow from the ocean boundary? That would seem to be something that could easily happen physically.

**Sec. 2.3** Maybe this comment isn't specific to this section, but generally there has been a fair bit of work in recent years on inferring the parameters of subglacial hydrologic models from indirect observations, and some of these would be worth citation, for example Irarrazaval, 2021 and Koziol, 2017 (and references therein).

**L162** I don't understand the strategy of holding either $W_r$ or $c_s$ constant while varying the other, which seems quite ad hoc. Why not do a proper grid search or pseudorandom sample?

**L169** This is a bit of a weird sentence: stability should not, a priori, influence experimental design. Of course, there is a de facto dependence when simulations don't converge, but this shouldn't enter the reasoning when designing which experiments to run.

**L169–180** I don't understand these criteria for steady state and why they are used for filtering simulations, and this section should be developed more substantially. The inputs to this model are not time-dependent, so why is there a problem for achieving steady state with respect to either flux or effective pressure? Why not just run longer? Does a lack of steady-state indicate numerical instability? How do you know that it's an instability and not a limit cycle? Why are simulations that achieve the flux steady state somehow more reliable than the other?

**Eq. 12** A single uniform $W_r$ is a nice numerical parameterization, but it seems unlikely that such a thing acutally exists, and that real bump heights are randomly distributed. This deserves a comment, because this notion of simulated specularity would seem to rely on actual uniformity in $W_r$.

**L205** I am quite skeptical of the justification for trying many combinations of model and data specularity thresholds as capturing transitional behaviors. It seems like a more reasonable justification is that it's not immediately obvious how to compare these proxies for subglacial conditions, and by doing a grid search for high correlations, you're doing a sort of ad hoc maximum likelihood estimator for the parameters of the model relating the two.

**L232** 'This range ...' which range is being referred to?

**Sec. 3.1.1** I think that reporting the ranges of parameters independently that agree with data (i.e. a marginal posterior distribution) is of limited utility because of the high likelihood of posterior correlations (e.g. only simulations that have a high $k_Q$ and low $k_q$ or vice versa might make it through the filter. Such a pattern has been seen in other works that infer parameters of subglacial hydrologic models). I think that a useful figure would be a plot where all four parameters are plotted pairwise, with both initial and filtered simulations presented (but colored differently). This would elucidate (at least) pairwise posterior dependencies between parameters. This is already done a bit in Figure 3, but I think it would be useful to extend it to other parameters as well.

**L315** Here and elsewhere, it would be helpful to define what is meant by 'below capacity'.

**L327** It would be helpful to see a figure illustrating this lake filling a draining. I don't actually see much of a mechanism in the model that allows for subglacial lakes to form to begin with: what does this look like in the model, and are its physics really capable of simulating such a thing?

**References**

[1] Irarrazaval, I., Werder, M., Huss, M., Herman, F., Mariethoz, G. (2021). Determining the evolution of an alpine glacier drainage system by solving inverse problems. Journal of Glaciology, 67(263), 421-434. doi:10.1017/jog.2020.116

[2] Koziol CP and Arnold N(2017) Incorporating modelled subglacial hydrologyinto inversions for basal drag.The Cryosphere11(6), 2783–2797. doi:10.5194/tc-11-2783-2017.

---

## Referee Comment (RC3)

**Reviewer comments on "Persistent, Extensive Channelized Drainage Modeled Beneath Thwaites Glacier, West Antarctica"**

**General comments**

This paper supports the existence of stable subglacial channels beneath Thwaites Glacier, and suggests that existing observations are incompatible with a distributed-only drainage system. The authors generate an ensemble of simulation results by sweeping through plausible parameter values, and then filter out results that are incompatible with both observed data as well as a number of physical constraints.

The configuration of the subglacial drainage system has consequences for drainage efficiency (and water pressures), submarine melting, and basal friction. Understanding each of these processes is vital if we are to understand the future evolution of Thwaites Glacier. This make the work presented in the paper particularly important.

A key part of this paper is the discussion of a methodology used to select data-compatible parameter values and subsequently drainage configurations. Generally, I think the method employed is sensible, but my general comments are on the consequences of some of the decisions made in this method.

How much of the observed behaviour is imposed (compared to emerging from the results) by the matching criteria? For example, criterion 1 (Line 212) compares zones, thus introducing a "special" line in between the lower and upper specular zones (zones 2 and 3) at the transition zone of Schroeder et al. 2013. Given that zone classification is discrete, it seems like any zone transition is likely to mark/impose a transition in the mode of drainage. Have some of the conclusions (such as the transition between drainage modes) been imposed based on the choice of selection criteria?

To give another example, if the specularity is a strong indicator of channelisation, then some of the observations in the selected runs will necessarily match with the specularity; namely, the extent of the channels. This is not a criticism of the criteria or methodology, but rather I think it would be good to distinguish the observations that can be directly inferred using the imposed data and criteria from those that emerge by incorporating the model. For the latter, I think insight about the nature of the channels (i.e. the number and size of channels) and the set of parameters values compatible with the observations demonstrate the benefit of using this methodology to interpret observations.

To finish, I wonder how a less-discrete compatibility criteria would compare to this method. For example, if you were to use the L2 error between the normalised $S$ and $R_{wt}$ fields. I imagine that this would resemble criteria 2, but would not require choosing critical thresholds. I am not suggesting the authors include this at all in their paper, I am just making a general comment.

In summary, the authors present a sensible methodology for making inference from some observed data (in conjunction with other physical constraints). As a consequence, they suggest that there may be significant channelisation beneath Thwaites Glacier. The existence of stable channels beneath Thwaites will have significant impact on the future of the glacier.

Overall I thought this paper was well written, and the conclusions well reasoned.

**Specific comments**

1) Given the importance of correlation as a measure of similarity I think it is important to say exactly how the correlation between masks of $R_{wt}$ and S is calculated.

2) In 3.1.1 you state how many runs remained after eliminating unsteady runs that don't satisfy criteria 1 and 2. However, for the remainder of the paper you only use runs data compatible runs, which also have sufficient water pressure. Did this additional criteria eliminate any of the 20/14 steady state runs that satisfy the comparison criteria? If so, it would be interesting to state here how many of your runs in total were data compatible. If not, is this condition (water pressure) at all necessary to include?

3) Section 2.4 was a bit unclear. It wasn't until the start of 3.1.1 that I knew how the criteria were applied. I think the second half of 2.4 should be re-thought to clarify the methodology. Particularly because I think the methodology is key to this paper. I think it is important to highlight that for each simulation there is 66 specularity—$R_{wt}$ combinations to compare and that if one of these combinations satisfy the criteria then the simulation is deemed realistic. I think Lines 226 and 227 say what needs to be said at the end of 2.4 (rather than line 215 which is too vague).

4) Did the specularity—$R_{wt}$ combinations suggest any particular, consistent values of critical S or $R_{wt}$ ? Presumable $S^{crt}$ is an important parameter by which we can interpret specularity results?

**Technical corrections**

In equation 6: is $\mathbf{q_c}$ a scalar value? If so, it would be clearer if it was not in bold. And if so, how is it calculated? Discharge is a vector so it isn't clear what the "discharge in the distributed system within a distance $l_c$" means. Presumable there it involves some integration of a dot product taken with respect to a direction. (If it is a vector, the absolute value of a vector should probably be clarified to mean the $L_2$ norm of the vector.) As it stands, more information about $\mathbf{q_c}$ is required to understand equation (6).

Line 117 : surfface -> surface

Figure 2 is first referenced on line 206. The caption for Figure 2 refers to FSS (flux steady-state) before this abbreviation is introduced in the text. It is not until line 228 that that FSS is defined to mean flux steady-state. Maybe just say flux steady-state rather than FSS in the caption of figure 2?

---

## Author Response (AR1)

**Dear Dr. Kang and Reviewers,**

We are very appreciative of your time in providing thoughtful comments and feedback for improving our manuscript. Overall, the reviews and assessment were positive. The main changes to the manuscript in response to these reviews were: 1) The expansion of Sections 2.3 and 2.4 to provide more detail about our parameter sweep, steady-state criteria, and specularity comparison process, 2) the addition of two appendices that further describe the topics covered in Sections 2.3 and 2.4, including additional supporting figures, 3) a new figure, Figure 1, that depicts important model inputs, and 4) minor wording changes and clarifications throughout the text to improve its readability. We hope these changes satisfy the concerns of the reviewers. In this document, reviews are given as bold italicized text, responses are provided in black text, and the changes to the manuscript are described in purple text. References to specific lines and figures in the manuscript refer to the revised text, unless stated otherwise. All changes are highlighted in the 'track changes' version of the manuscript. Again, we thank you for your time and careful reading of our manuscript.

Sincerely,

**Alex Hager**

*Review of "Persistent, Extensive Channelized Drainage Modeled Beneath Thwaites Glacier, West Antarctica - Hager et al."*

**Summary of the paper:**

The manuscript investigates the possibility of formation of channelized drainage under the Thwaites glacier using subglacial hydrology component of MPAS-Albany Land Ice model. Earlier, Schroeder et al. 2013 reported transition of water system from distributed to efficient drainage system in the interior of Thwaites glacier from specularity content data of bed echo. In this paper, the authors run several numbers of simulations by varying four parameters (sheet conductivity, channel conductivity parameter, Cavity spacing, and bed bump height) and compares the modelled results with the specularity content data of Schroeder et al. 2013 to assess the likelihood of channelization under Thwaites glacier.

The authors discuss their results with different geophysical properties of the Thwaites glaciers, some of which are important for further investigation. A counter-example analysis of channelization is also provided by running some simulations with channelization disabled which is a great addition. It is subject to discussion whether the findings are robust, but the presented results are very well complemented with counter examples, available data and adequate discussions, and that makes it worth getting published.

The paper is overall well written. The introduction nicely covers previous knowledge of subglacial hydrology in general and this region in particular. The results are very well- articulated. The authors complement their results with other observations such as ice shelf basal melt and present some interesting discussions about the findings of the paper, such as effective pressure consideration at the grounding line in sliding laws, etc., which is important for future research. Overall various geophysical aspects of Thwaites glacier are discussed under the findings of this paper, and all these make the content

**of the paper rich. However, the paper has some loose ends which requires some work. I mention them in the following:**

We thank the reviewer for their thorough review and for providing useful suggestions for improving the manuscript. We have addressed all concerns or provided further explanations to our choices of methodology. As most of the reviewer's concerns regarded our method for comparison between specularity content and model output, we have provided detailed additional information explaining our choice of comparison method that we hope will be satisfactory to the reviewer.

**Comments:**

1. The method section, especially section 2.4 needs substantial work. This part is very much unclear. The physical basis of choosing the specularity content threshold is not clear to me. Similarly, it is not clear why 6 different Rwt threshold values are used instead of one Rwt (e.g. 0.95). At present it is not easy to understand how different simulations are done with different combinations of parameters. I would suggest to provide a table explaining the simulations. Additionally, it is not obvious how different parameter sweep leads to certain number of simulations.

For each set of bed roughness parameter combinations, we sampled and expanded the conductivity parameter space at consistent intervals until runs either failed to reach steady-state or had average water pressures below 90% flotation (lines 163–165). Useable conductivity parameter space varied with different sets of bed roughness parameters, so we were not able to conduct the same number of runs for every set of bed roughness parameters. The authors would be open to including a table of all of our runs in a supplement, or conversely, include supplementary figures analogous to Figure 3 for all 6 sets of roughness parameters (see response to RC2, comment on Section 3.1.1). These figures would similarly illustrate average water pressures, correspondence to specularity content, and which runs reached what steady-state criteria. Please see response to comment #3 for a discussion on specularity content and Rwt thresholds.

Two paragraphs were added at Lines 227–243 that explain our choices of specularity and Rwt thresholds (see response to comment #3) for creation of the binary masks. We also made slight wording changes at Lines 166–167 and 169–172, as well as added Appendix A, to better explain our choices of bed roughness parameters and the reason for the differing numbers of runs per combination of these parameters. Appendix A includes new figures (A1–A5), akin to Figure 4, that depict the conductivity parameter sweep for each set of bed roughness parameters.

2. Whereas the parameters choices are very well explained, the choice of thresholds lacks sufficient explanations. There are many thresholds used (flotation, flux steady state, Pressure steady state (all in sect. 2.3), Scrt, Rwt, correlation coefficient> 0.35 (sec. 2.4)) and results of this paper are highly dependent on the choices of these threshold. These choices remain very subjective, and not enough supported analysis is provided for their choice. I would recommend to provide substantial logic for using those thresholds and have detail discussion around them. The authors need to present some more statistics to support these choices, and it can be included as appendix or as supplementary, if not as main article. That brings me to the question that how does your result are sensitive to the choice of threshold?

Establishing steady state criteria inherently involves defining a cutoff threshold for acceptable noise remaining in the model. For our pressure-steady runs, effective pressure at each cell is allowed to

fluctuate 0.5% of its value on average. This equates to an allowable fluctuation of roughly 1 kPa where effective pressure is lowest (~200 kPa) and 10 kPa where effective pressure is highest (~2000 kPa). For flux steady-state runs, meltwater production above each transect must equal the total discharge across the transect within 0.5%. Total melt production above the grounding line is roughly 155 m3/s, so our steady-state criteria require that we know the total grounding line discharge within 0.8 m3/s. Given this analysis, we believe the choices of steady-state thresholds are fairly strict and do not meaningfully influence our results. No data-compatible runs had average water pressures below 91% flotation, so our requirement that acceptable runs have water pressures >90% does not influence our results (please see response to specific comment #2). Please see response to the following comment for discussion about the thresholds used in comparison between specularity content and  $R_{wt}$ .

Lines 538–543 were added to Appendix A describing the sensitivity of our results to our steady-state criteria. A reference to Appendix A was added to section 2.3 at Lines 196–197. Please see responses to the preceding and succeeding comments about changes regarding other thresholds.

3. Comparison with specularity content seemed bit like cherry-picking. However, I do not deny the potential of specularity data in understanding subglacial hydrology. I just feel that these data can be used better/sophisticated way to infer status of subglacial hydrology. Results associated with specularity content are not very robust and presented in very sporadic manner. In my opinion, this is the major area of improvement for the manuscript. The authors should explore better way to have comparison with specularity content data. The choice of threshold of specularity data is not clear. There is no physical basis of it. Furthermore, the description needs to be improved substantially. At present, this part is not completely clear to me. I would recommend to add more detail description with figures for this section to enhance the readability.

**Although I do not have any clear suggestions on specularity content data, but the authors should find a better way to compare the specularity data which I think does not require any additional model runs. The present way of representation and analysis is neither very convincing, nor easy to understand.**

High specularity content and high  $R_{wt}$  both unequivocally represent broad, flat areas of pooled water, yet the two are governed by independent processes and likely do not covary when their values are low. This makes comparing the two difficult, and a simple spatial correlation unlikely to work as a comparison method. Comparisons between the two quantities should instead rely on spatial point patterns (such as our binary masks) that map where specularity content and  $R_{wt}$  are high. Unfortunately, this method does require choosing critical thresholds of what is considered "high" for each quantity. We address this problem by creating a population of masks for each variable, each using a different critical threshold within a reasonable range (see below for determination of "reasonable values"), and comparing all 66 combinations of specularity content and  $R_{wt}$  masks. Data-compatible runs only have to match one mask combination, which makes our comparison less sensitive to our choices of critical thresholds.

Absolute values of specularity depend on the geometry of ice thickness, survey geometry, radar processing, and subglacial water geometry (Schroeder et al., 2013; Schroeder et al., 2015; Young et al., 2016; Haynes et al., 2018). As a result, the relative specularity can be interpreted as a measure of the relative "amount of bed covered by flat subglacial water bodies" within the glacier. For this we set a threshold based on the cumulative distribution of specularity within the particular survey/glacier (Figure RC1a). Comparison with the model is sensitive to the choice of threshold, so the comparison is repeated using a range of different thresholds between 0.15 - 0.25, which selects for the greatest  $\sim 5 - 15\%$  of our specular data. This is a conservative and deliberatively empirical approach focused on comparing the

water transition expressed in the specularity in Schroeder et al. (2013) with our modeling. There is potential for finer-scale local analysis of specularity signals by adapting and expanding the electromagnetic modeling approach in Mark Haynes et al. (2018) to the glacier-catchment scale, which is exciting but beyond the scope of this paper (which is focused on investigating and understanding transitions in subglacial system configuration in ice sheet modeling and comparing that to broad-scale patterns in specularity data).

As with our choice of Scrt, there is a range of  $R_{wt}$  that could simulate specularity in our model (this occurs near  $R_{wt} = 1.0$ ), and our comparison results are sensitive to this choice. To address this, we again create multiple masks of  $R_{wt}$  using different thresholds ranging between 0.95 and 1.0 and require only one of these masks match one specularity mask. This approach allows us to minimize the sensitivity of our analysis to our choices of thresholds.

Matching specularity and Rwt masks is essentially a comparison between two spatial point patterns. Such a comparison is challenging as it requires a global statistic that can recognize local patterns of point clusters. We explored many alternative methods of mask comparison, including calculating the spatial similarity index described in Andresen (2009; 2014), which segments the domain into multiple areal units and uses a Monte Carlo approach to determine an overall similarity statistic across all areal units. However, this method is highly sensitive to the size, shape, and location of areal units, and was largely unsuccessful at identifying similar specularity and Rwt masks.

The method developed in the current paper shares the concept of areal units by defining four physicallybased zones within which we assess similarity between the two masks. These zones are intentionally chosen to loosely encompass regions of specularity or non-specularity, which allows for some spatial variability between masks and decreases the sensitivity to the zonal boundaries. We then require the two specularity and Rwt masks to match at 50% or more of grid points within each zone. While this criterion does well by itself in selecting positive matches, it also selects many false positives. This occurs when the Rwt mask is almost entirely non-specular and over 50% of the cells in each zone is non-specular in the specularity mask. It is therefore necessary to include a second criterion that can remove these false positives, which we do by requiring an overall correlation coefficient of  $r \ge 0.35$ . Again, correlation by itself does a fair job at identifying positive matches, but it also identifies false positives when the Rwt mask is overly specular. As the two criteria fail for opposing reasons, they can check and balance each other if the thresholds are tuned appropriately. We acknowledge this comparison method is sensitive to multiple choices of thresholds, so we attempt to make our criteria for selecting data-compatible runs as generous and inclusive as possible while still removing runs that clearly do a poor job at resembling observations. We empirically determined that requiring  $\geq$  50% of cells in each zone to agree and r  $\geq$ 0.35 works well at identifying positive matches and is sufficiently general to allow a reasonable variety of Rwt masks to pass this filtering process.

The authors acknowledge that some of the above information was not included Section 2.4, yet would be helpful in making a convincing argument for our comparison method. We will adjust section 2.4 or provide a supplement to better explain the justification for the two criteria and how they complement each other. We can also provide supplementary figures illustrating various combinations of specularity and Rwt masks together with their comparison statistics (ie. Figures 2b, 2d) so that the reader can see how our two criteria identify positive and negative matches.

**Figure RC1a**: Histogram of the cumulative density function of specularity content from Schroeder et al. (2013). Red band indicates the range of Scrt thresholds used for binary masks.

As mentioned above, more information regarding our choices of specularity and Rwt thresholds have been added to Section 2.4 at Lines 227–243. In addition, we also added Appendix B, which provides further information about our comparison method (most of which is included above). Figure B1 depicts multiple mask comparisons for one data-compatible model simulation, along with statistics showing which masks combinations were successful matches and why. We hope this figure, together with the additional information provided in Section 2.4 and Appendix B, satisfies the reviewer's concerns.

In addition, I have some major specific questions/comments:

1. Can you please elaborate why two different steady-states criteria are chosen?

Please see response to RC2, comment L169–180.

Lines 176–180, 184–186, and 192–197 have been added or reworded to better explain the need for two different steady-state criteria.

2. Please provide justification of considering avg. water pressures of >90% flotation. Using 90% only is very subjective. It would be good to provide supporting result of choosing 90%. For example, show how your results will differ when using 80% or 95% flotation.

The standard of  $\geq$  90 % flotation was used as a first-order criterion for determining when runs may be unrealistic and when to stop expanding parameter space. This choice is consistent with observed and modeled water pressures near flotation observed at Ice Stream B (Engelhardt and Kamb, 1997) and Pine Island Glacier (Gillet-Chaulet et al., 2016), respectively. We built upon this constraint by imposing our steady-state criteria and specularity comparison standards. No data-compatible runs had average water pressures below 91% flotation, so this choice does not affect our results. We acknowledge the wording surrounding this criterion is confusing, and it will be addressed in a later version of the manuscript.

Lines 158–162 were reworded to provide a clearer justification for this criterion of when to stop expanding parameter space. Additionally, the sentence "Simulations that did not reach either steady-state criteria, or had water pressures <90% flotation, were discarded" was shortened to omit the phrase regarding water pressure (Line 196). This was done in accordance with Reviewer 3's Specific Comment #3, who pointed out that this was a redundant criterion for data-compatible runs, as no data-compatible runs had water pressures below 91%.

3. The effective pressure (N) in the interior seems bit high especially where specularity content is high which is supposed to represent distributed drainage! Can you have some discussions on your derived effective pressure value with effective pressure reported in other studies? You provide good discussion with the discharge from previous studies of Antarctica, but I would recommend to do the same for effective pressure.

Modeled effective pressure is lowest where specularity content is high, which is in agreement with the presence of a distributed system at these locations. One of the novelties of this study is that it provides estimates of effective pressure beneath part of the West Antarctic Ice Sheet, which is largely unknown to date. To the authors' knowledge, the only direct observations of effective pressure in West Antarctica were measured via a borehole at Ice Stream B (Engelhardt and Kamb, 1997). Those authors report effective pressure ranging between -30 – 160 kPa, similar to average effective pressures of 200 – 500 kPa in the highly specular zones in our data-compatible runs. As the observations of Engelhardt and Kamb (1997) were measured at one point in an ice stream, we would expect their reported effective pressures to be less than those at upper Thwaites Glacier, where ice flow is slower. We can incorporate this comparison into section 4.2.2, as well as any other suggestions of observational constraints of effective pressure in West Antarctica.

After further deliberation, the authors have decided that including a direct comparison between effective pressures in our model and those measured at Ice Stream B may not be appropriate. The borehole observations were collected at one point within an ice stream, and it is likely that effective pressures measured there may only be roughly similar to those at Thwaites Glacier, where different basal processes are occurring. We therefore opt to keep the comparison as a general statement regarding water pressures "near flotation," which is included in Lines 160–162 (Section 2.3).

4. In addition to above comment on sec. 2.4, I would recommend to add few figures showing different masks derived using thresholds of Scrt or Rwt. If not here, these figures must be provided in appendix or in supplementary.

Please see response to above comment on section 2.4.

Figure B1 has been added to the manuscript and addresses the reviewer's concerns.

5. The paper presents nice analysis with parameter sweep and completement their results with ice shelf basal melt rate. I think this part can get more focus as it is an interesting comparison (e.g., Wei et al., 2020).

Section 4.2.1 is devoted to the impact of subglacial channels on ice shelf basal melt rates. A comparison to Wei et al. (2020) would fit nicely in this section and will be included in a later version of the manuscript.

A sentence referencing Wei et al. (2020) has been added at Lines 388–389.

**Minor comments**

Line 215: 'majority of the cells' - How many number or cells do you mean here?

Greater than 50% of the cells within each zone had to agree between the specularity and  $R_{wt}$  masks. The exact number of cells varies by zone, as each zone has a different number of cells.

This is clarified in Line 551 in Appendix B.

*L* 229-230: Was zero instances of water pressure below 90% outside data-compatible run? I would suggest to include a table here too with these criteria. Otherwise, this line remains bit vague.

Please see above response to major specific comment #2

Figures 4 and A1–A5 now depict binned average water pressures for every run, along with which runs were data-compatible. No data-compatible runs had water pressures below 90%.

*L* 256- 257: ".. the 50–100 km transition to channelized flow coincides with a substantial increase in basal friction melt rate." - Can you please elaborate on that with some data?

A new figure depicting the sliding velocity, basal traction, and basal friction melting can be added to the methods section of the paper so the reader can compare the primary model forcing to the results in Figure 4.

Figure 1 was added to the manuscript, which depicts bed topography, basal sliding speed, basal friction heat flux, and the production of basal meltwater used as inputs in the model. 50 km contour lines were added to each subplot so that the reader can easily compare to the channelization plots in Figures 5 and 6.

*Figures 2,6 - What does the black dots in b c3and d represents? Are these the locations of significant correlations?*

The black dots in Figures 2 and 6 are the binary masks of specularity content and  $R_{wt}$ , as described in section 2.4 and in the figure captions. The authors are open to wording suggestions to make the captions clearer to the reader.

Minor wording changes were made to the captions of these figures (Figures 3 and 7 in the revised manuscript) to improve clarity.

Fig. 5: Does '> 5m2s-1' include >10 m3s-1? or it is >5 and <10?

 $Q > 5 \text{ m}^3/\text{s}$  includes  $Q > 10 \text{ m}^3/\text{s}$ , and does not mean  $5 - 10 \text{ m}^3/\text{s}$ .

The authors feel this is adequately defined in the text and did not make changes to the manuscript here.

L 275 -277: ".. pressures near the upper domain boundary, although effective pressures within 300 km of the terminus are in strong agreement with the low-resolution model." - It is not clear to me from the figure.

The comparison here is between Figure 4b and Figure 4c. The authors feel these subplots accurately depict the sentence in question, but are open to suggestions about how to make this more clear to the reader. One possibility would be to add a centerline transect to Figure 7a from the high-resolution model.

Centerline transect effective pressures from the high-resolution run was added to Figure 8a (formerly Figure 7a). A reference to this figure, as well as Figure 5 (formerly Figure 4), was added as Line 315.

L561: The author list is incomplete.

This will be fixed in the next version of the manuscript

Fixed in manuscript (Line 647).

**REFERENCES:**

Andresen, M. A.: Testing for similarity in area-based spatial patterns: A nonparametric Monte Carlo approach, Applied Geography, 29, 33–345, doi:10.1016/j.apgeog.2008.12.004, 2009

Andresen, M. A.: An area-based nonparametric spatial point pattern test: The test, its applications, and the future, Methodological Innovations, 9, 1–11, doi: 10.1177/2059799116630659, 2014

Engelhardt, H., Kamb, B.: Basal hydraulic system of a West Antarctic ice stream: constraints from borehole observations, Journal of Glaciology, 43, 144, 207–230, 1997

Gillet-Chaulet F., Durand G., Gagliardini O., Mosbeux C., Mouginot J., Rémy F., and Ritz C.: Assimilation of surface velocities acquired between 1996 and 2010 to constrain the form of the basal friction law under Pine Island Glacier, Geophyscial Research Letters, *43*, 10,311–10,321, doi:10.1002/2016GL069937, 2016

Haynes, M. S., Chapin, E., Schroeder, D. M.: Geometric power fall-off in radar sounding, IEEE Transactions on Geoscience and Remote Sensing, 56, 11, 6571–6585, doi: 10.1109/TGRS.2018.2840511, 2018 Schroeder, D. M., Blankenship, D. D., and Young, D. A.: Evidence for a water system transition beneath Thwaites Glacier, West Antarctica., Proceedings of the National Academy of Sciences of the United States of America, 110, 12225–8, https://doi.org/10.1073/pnas.1302828110, 2013

Schroeder, D. M., Blankenship, D. D., Raney, R. K., and Grima, C.: Estimating subglacial water geometry using radar bed echo specularity: application to Thwaites Glacier, West Antarctica, IEEE Geoscience and Remote Sensing Letters, 12, 443–447, 2015

Young, D., Schroeder, D., Blankenship, D., Kempf, S. D., and Quartini, E.: The distribution of basal water between Antarctic subglacial lakes from radar sounding, Philosophical Transactions of the Royal Society A: Mathematical, Physical and Engineering Sciences, 374, 20140 297, 2016

*Referee comment on 'Persistent, Extensive Channelized Drainage Modeled Beneath Thwaites Glacier, West Antarctica'*

Doug Brinkerhoff January 2022

**1 Summary**

In this section, the authors make the case for significant channelization in the Thwaites catchment, which may have substantial implications for its evolution due to both the modification of sub-shelf circulation by melt plumes and also a relatively higher upstream basal shear stress due to efficient evacuation of water (and thus a reduction in water pressure). This argument primarily relies upon the comparison of a numerical model of drainage development with putative radar-based observations of a proxy for drainage type. A moderately large ensemble of model configurations are tested, leading to several configurations that are consistent with observations, all of which involve the development of substantial channelization.

I find this paper to be well written and compelling. I particularly appreciate the ensemble approach to the modelling, which greatly improves the robustness of the results. I do not have any major comments; however I do have a handful of minor comments, clarifications, and technical corrections that I would like to see addressed.

We thank Dr. Brinkerhoff for his thoughtful and detailed review our paper, and for providing useful suggestions for improving our manuscript. We have addressed all concerns and have provided clarifying explanations where necessary.

Line-by-line comments

L92 I suggest mentioning that the implications of neglecting the pressure change term are discussed in the discussion.

A sentence mentioning this will be added to the next draft of the manuscript.

This has been added to Lines 94–95 in the revised text.

**Eq. 7 Where are the channels? Are they defined on the edges of the Voronoi mesh, or between center points, or something else entirely?**

Channels segments connect the center points of neighboring cells, and channel velocities and fluxes are computing at the edge midpoints between two cells. A few sentences describing where variables are computed on the Voronoi mesh can be added to the manuscript, although this has been done in detail in Hoffman et al. (2018).

Lines 64–66 have been added defining where these variables are calculated on the model mesh.

L101 I suggest language like 'time derivatives are discretized using an explicit forward Euler method ...'.

This language will be revised in the manuscript to improve clarity.

This sentence has been changed in Line 104 in accordance with the reviewer's suggestion.

L104 What are these 'transients' mentioned here? Numerical instabilities? Usually this word refers to non-physical changes induced by initial conditions that are far out of balance with the governing equations, in which case, temporal averaging wouldn't seem to do much.

The "transients" described here are minor, stable oscillations similar to the limit cycles described in Kingslake (2014) and Schoof (2020), and result from pressure fluctuations caused by the englacial storage term or bedrock cavity space. Taken as a multi-year average, these small perturbations fluctuate around a steady-state, and are not consequential to our model results. The authors are open to any language changes here if it would improve clarity.

"Transients" has been changed to "oscillations" to avoid any ambiguity in this sentence (Line 107).

**L117 'surfface' $\rightarrow$ 'surface'**

This will be changed in the next version of the manuscript.

Fixed in manuscript (Line 120).

**L123 I don't understand the notion of varying hydropotential values: isn't sea level just sea level? Depth wouldn't have anything to do with it.**

Hydropotential in ocean cells is calculated as  $\phi_o = \rho_w g Z_b - \rho_o g Z_b$ , where  $\rho_w$  is the density of fresh water, g is gravity,  $Z_b$  is the bed elevation, positive above sea level, and  $\rho_o$  is the density of ocean water. Thus, the freshwater subglacial hydrologic system at the bed 'feels' the hydrostatic pressure of the overlying ocean water column at the grounding line boundary, which varies spatially with changes in bathymetry.

The authors feel this is adequately described in the text and have not made any changes to the manuscript here.

**L123 Why suppress backflow from the ocean boundary? That would seem to be something that could easily happen physically.**

Backflow into the subglacial system from the ocean is theoretically likely to occur up to several kilometers inland from the grounding line in both the distributed and channelized systems (Wilson et al., 2020; Robel et al., 2022). However, the theory describing this process is not fully developed, nor is it currently implemented in two-dimensional subglacial hydrology models. Saltwater intrusions are thought to propagate into the subglacial system as a salt wedge beneath outflowing subglacial freshwater that is likely influenced by a combination of turbulent mixing, double diffusive instabilities, tidal pumping, spatially-varying bed lithologies, and other complicating factors (Wilson et al., 2020; Robel et al., 2022). Without the ability to resolve this complex two-layer exchange flow, our model can form unrealistic instabilities near the grounding line akin to a Jökulhlaup draining from the ocean into the glacier interior when effective pressure gets very low. It is rare in the model, but due to fixed sea level, when it does occur, it results in unchecked exponential channel growth and instability.

Suppressing backflow from the ocean is therefore the most realistic choice for our model; however, we acknowledge this as a potential limitation and future avenue for model improvement.

The word "unstable" was inserted into this sentence (Line 130) to clarify that this is an artificial instability in the model and is not representative of naturally occurring inflow from the ocean.

Sec. 2.3 Maybe this comment isn't specific to this section, but generally there has been a fair bit of work in recent years on inferring the parameters of subglacial hydrologic models from indirect observations, and some of these would be worth citation, for example Irarrazaval, 2021 and Koziol, 2017 (and references therein).

The next version of the manuscript will reference other studies, such as those provided by the reviewer, that used alternate approaches to infer unknown subglacial hydrology parameters.

Lines 136–137 was added acknowledging previous studies that inferred hydraulic parameters using inversion techniques. The wording in the subsequent sentences (Lines 137–140) was altered to improve readability following the insertion of Lines 136–137.

**L162 I don't understand the strategy of holding either Wr or cs constant while varying the other, which seems quite ad hoc. Why not do a proper grid search or pseudorandom sample?**

While a full sweep of the conductivity parameter space is done for each bed roughness combination, the determination of bed roughness parameters is closer to a sensitivity analysis (varying one parameter at a time). This is done intentionally in part to decrease the number of necessary model runs, and to ease the complexity of a four-dimensional parameter sweep. This choice is made because real physical constraints exist for conductivity parameters (lines 136 – 156), while bed roughness parameters are theoretical quantities approximating the broad, general characteristics of the bed that only have indirect physical corollaries. Therefore, it makes sense to explore all of realistic parameter space for conductivity values, yet undertake a sensitivity analysis (deviating from typical values in literature) for bed roughness parameters that lack real physical constraints. We acknowledge our approach is a bit ad hoc and lacks the rigor of a proper uncertainty quantification.

A slight wording change was made in Lines 166–167 to emphasize the sensitivity analysis conducted for bed roughness parameter. A more detailed explanation of our parameter sampling methodology is then provided in Lines 5301–537 of Appendix A, along with the addition of Figures A1–A5.

L169 This is a bit of a weird sentence: stability should not, a priori, influence experimental design. Of course, there is a de facto dependence when simulations don't converge, but this shouldn't enter the reasoning when designing which experiments to run.

Please see response to next comment.

See changes made for next comment.

L169–180 I don't understand these criteria for steady state and why they are used for filtering simulations, and this section should be developed more substantially. The inputs to this model are not time-dependent, so why is there a problem for achieving steady state with respect to either flux or

effective pressure? Why not just run longer? Does a lack of steady-state indicate numerical instability? How do you know that it's an instability and not a limit cycle? Why are simulations that achieve the flux steady state somehow more reliable than the other?

Runs failed to reach steady state for two main reasons: 1) A local numerical instability developed in the channel model, or 2) the domain became over-pressurized so that the adaptive timestep failed to progress. Instabilities occurred when discharge in a channel was unrealistically high (normally this occurred because the channel conductivity was too high or the distributed conductivity was too low) so that an unstable feedback cycle developed where unrealistic channel discharge increased channel dissipative melting, which in turn led to higher channel discharge and more dissipative melting. This normally occurred locally at a few neighboring grid cells, so that the water pressure and thickness elsewhere in the domain was unaffected. Hence, we developed the pressure steady state criteria so that these runs could still yield some useful information about aspects of the subglacial drainage system other than discharges. In some cases, instabilities could be avoided by changing the englacial porosity, which acts as a buffer between meltwater production and the subglacial system but does not affect the steady state configuration.

Runs became over-pressurized at very low distributed conductivities where the distributed system could not compensate the net input of meltwater, and the adaptive timestep became impractically small to meet the pressure CFL condition. Runs that failed to reach flux steady-state did not represent steady systems where the subglacial discharge realistically balanced the production of meltwater, and so it was not possible to accurately assess the relative fraction of channel discharge to distributed system discharge. As our goal was to explore as much of parameter space as possible, runs were continually restarted until either reaching flux steady state, forming an unpreventable numerical instability, or becoming computationally untenable to keep running. A more detailed description of this process and the reasoning behind it can be added to section 2.3 in the manuscript or to a supplement.

**The above information has been added at Lines 176–180, 184–186, and 192–197.**

**Eq. 12 A single uniform $W_r$ is a nice numerical parameterization, but it seems unlikely that such a thing acutally exists, and that real bump heights are randomly distributed. This deserves a comment, because this notion of simulated specularity would seem to rely on actual uniformity in $W_r$ .**

The authors agree that a spatially uniform  $W_r$  likely does not actually exist, and that this deserves a comment here in the manuscript. It an underlying assumption of MALI and similar subglacial hydrology models (ie., Schoof, 2010; Hewitt, 2011; Werder et al., 2013) that this parameterization of bed roughness broadly captures sub-grid cell bed characteristics, and it remains the best approximation for bed properties *in lieu* of very high-resolution (< meter scale) observations of the bed beneath the Antarctic Ice Sheet. However, this does not change the reasoning behind Equation 12, which would still simulate specularity if  $W_r$  varied spatially.

Two sentences acknowledging the unrealistic nature of a uniform  $W_r$ , as well as the ability of  $R_{wt}$  to predict high specularity even in heterogeneous bed conditions were added at Lines 224–226.

L205 I am quite skeptical of the justification for trying many combinations of model and data specularity thresholds as capturing transitional behaviors. It seems like a more reasonable justification is that it's not immediately obvious how to compare these proxies for subglacial conditions, and by doing a grid search

**for high correlations, you're doing a sort of ad hoc maximum likelihood estimator for the parameters of the model relating the two.**

The transitional behavior is an important reason that comparison between modeled and observed specularity is not straightforward; however, the authors agree that the reviewer's justification for this comparison method is convincing. This section can be revised to combine both justifications, as well as address the concerns of the other reviewers' comments (see response to RC1, comment #3).

Our reasoning for using multiple binary masks has been expanded in accordance with Reviewer 1's concerns (Lines 227–243 in the text). The new wording focuses on building a population of mask combinations to account for the sensitivity of our results to our choices of thresholds. We believe the new reasoning is also aligned with Reviewer 2's comment.

**L232 'This range ...' which range is being referred to?**

This sentence is referring to the range of  $k_q$  values that were seen in data-compatible runs. This distinction will clarified in the next draft of the manuscript.

"This range" has been changed to "The range of  $k_q$  in data-compatible runs" (Line 271).

Sec. 3.1.1 I think that reporting the ranges of parameters independently that agree with data (i.e. a marginal posterior distribution) is of limited utility because of the high likelihood of posterior correlations (e.g. only simulations that have a high  $k_Q$  and low  $k_q$  or vice versa might make it through the filter. Such a pattern has been seen in other works that infer parameters of subglacial hydrologic models). I think that a useful figure would be a plot where all four parameters are plotted pairwise, with both initial and filtered simulations presented (but colored differently). This would elucidate (at least) pairwise posterior dependencies between parameters. This is already done a bit in Figure 3, but I think it would be useful to extend it to other parameters as well.

The authors agree this could be helpful; however, because we use consistent values for each parameter, the plots described by the reviewer would result in many overlapping points and would be difficult to interpret. Instead, we can include versions of Figure 3 for each bed roughness parameter combination in a supplement. We hope this will address the reviewer's concerns.

Figures A1–A5 have been added to Appendix A that depict the conductivity parameter sweep for each set of bed roughness parameters (in addition to Figure 4). We hope these additional figures satisfy the reviewer's concerns.

L315 Here and elsewhere, it would be helpful to define what is meant by 'below capacity'.

A definition of "capacity" can be included at line 193, where the word is first used.

A parenthetical definition of "below capacity" was inserted at Line 210-211, where the phrase is first used.

L327 It would be helpful to see a figure illustrating this lake filling a draining. I don't actually see much of a mechanism in the model that allows for subglacial lakes to form to begin with: what does this look like in the model, and are its physics really capable of simulating such a thing?

As briefly discussed in lines 330 – 335, our model lacks the proper physics to accurately capture realistic subglacial lake filling/draining behavior, and the behavior seen in our high-resolution is likely akin to the limit cycles of lake filling and draining described in Kingslake (2014) and Schoof (2020). A figure or video illustrating this behavior this could be added to a supplement. Conversely, the authors are also open to omitting this paragraph, as this discussion point does not impact our primary results and is still somewhat speculative.

This paragraph was removed from the text as it was too speculative and equivocal. In its place, we added a sentence at Lines 364–367 explaining that our model lacks the proper physics to investigate subglacial lake drainage, but that this could be an important avenue of future work.

**References**

[1] Irarrazaval, I., Werder, M., Huss, M., Herman, F., Mariethoz, G. (2021). Determining the evolution of an alpine glacier drainage system by solving inverse problems. Journal of Glaciology, 67(263), 421-434. doi:10.1017/jog.2020.116

[2] Koziol CP and Arnold N(2017) Incorporating modelled subglacial hydrology into inversions for basal drag. The Cryosphere11(6), 2783–2797. doi:10.5194/tc-11-2783-2017.

**ADDITIONAL REFERENCES PROVIDED BY AUTHORS:**

Hewitt, I. J.: Modelling distributed and channelized subglacial drainage: the spacing of channels, Journal of Glaciology, 57, 302–314, https://doi.org/10.3189/002214311796405951, 2011

Hoffman, M. J., Perego, M., Price, S. F., Lipscomb, W. H., Jacobsen, D., Tezaur, I., Salinger, A. G., Tuminaro, R., and Zhang, T.: MPAS- Albany Land Ice (MALI): A variable resolution ice sheet model for Earth system modeling using Voronoi grids, Geoscientific Model Development, pp. 1–47, https://doi.org/10.5194/gmd-2018-78, 2018

Kingslake, J.: Chaotic dynamics of a glaciohydraulic model, Journal of Glaciology, 61, 227, 493–501, doi: doi: 10.3189/2015JoG14J208, 2014

Robel, A., Wilson, E., Seroussi, H.; Layered seawater intrusion and melt under grounded ice, The Cryosphere, 16, 451–469, https://doi.org/10.5194/tc-16-451-2022, 2022

Schoof, C.: Ice-sheet acceleration driven by melt supply variability, Nature, 468, 803–806, https://doi.org/10.1038/nature09618, 2010

Schoof, C.: An analysis of instabilities and limit cycles in glacier-dammed reservoirs, The Cryosphere, 14, 3175–3194, https://doi.org/10.5194/tc-14-3175-2020, 2020

Werder, M. A., Hewitt, I. J., Schoof, C. G., and Flowers, G. E.: Modeling channelized and distributed subglacial drainage in two dimensions, Journal of Geophysical Research: Earth Surface, 118, https://doi.org/10.1002/jgrf.20146, 2013

Wilson, E., Wells, A. J., Hewitt, I. J., Cendese, C.: The dynamics of a subglacial salt wedge, Journal of Fluid Mechanics, 895, A20, doi:10.1017/jfm.2020.308, 2020

Reviewer comments on "Persistent, Extensive Channelized Drainage Modeled Beneath Thwaites Glacier, West Antarctica"

**General comments**

This paper supports the existence of stable subglacial channels beneath Thwaites Glacier, and suggests that existing observations are incompatible with a distributed-only drainage system. The authors generate an ensemble of simulation results by sweeping through plausible parameter values, and then filter out results that are incompatible with both observed data as well as a number of physical constraints.

The configuration of the subglacial drainage system has consequences for drainage efficiency (and water pressures), submarine melting, and basal friction. Understanding each of these processes is vital if we are to understand the future evolution of Thwaites Glacier. This makes the work presented in the paper particularly important.

We thank the reviewer for their thorough review and for providing useful suggestions for improving the manuscript. We have addressed all their concerns and have provided additional material and clarification about our choice of comparison criteria between observed specularity content and our model output.

A key part of this paper is the discussion of a methodology used to select data- compatible parameter values and subsequently drainage configurations. Generally, I think the method employed is sensible, but my general comments are on the consequences of some of the decisions made in this method.

How much of the observed behaviour is imposed (compared to emerging from the results) by the matching criteria? For example, criterion 1 (Line 212) compares zones, thus introducing a "special" line in between the lower and upper specular zones (zones 2 and 3) at the transition zone of Schroeder et al. 2013. Given that zone classification is discrete, it seems like any zone transition is likely to mark/impose a transition in the mode of drainage. Have some of the conclusions (such as the transition between drainage modes) been imposed based on the choice of selection criteria?

The goal of our comparison is to determine which model runs simulate the pattern of specularity content observed by Schroeder et al. (2013), which is used to infer channelization near the terminus and is not by itself proof of a drainage style transition. For the comparison, we define four zones based on the presence or absence of observed specularity content that

data-compatible runs should be able to reproduce. Of these four zones, only the border between Zone 2 (specular) and Zone 1 (non-specular) could represent a change in dominant drainage style; however, it is not clear why this occurs if only relying on observed specularity content. The transition to the non-specular Zone 1 could occur for a variety of reasons, including: 1) widespread convex channels refracting radar energy across the entire glacier width (as hypothesized by Schroeder et al., 2013), 2) a few discrete channels removing water from the surrounding distributed system (as supported by our modeling), or 3) a distributed system that is below capacity due to supercooling or other reasons. Therefore, our comparison criteria do not pre-determine a model outcome, but allows us to test which drainage configuration(s) can explain the pattern of observed specularity content. Please refer to our response to RC1, comment #3, for additional information about why we chose our two comparison criteria, and how they work in tandem to select for data-compatible model runs.

**The above information has been consolidated in Lines 545–554 in Appendix B.**

To give another example, if the specularity is a strong indicator of channelisation, then some of the observations in the selected runs will necessarily match with the specularity; namely, the extent of the channels. This is not a criticism of the criteria or methodology, but rather I think it would be good to distinguish the observations that can be directly inferred using the imposed data and criteria from those that emerge by incorporating the model. For the latter, I think insight about the nature of the channels (i.e. the number and size of channels) and the set of parameters values compatible with the observations demonstrate the benefit of using this methodology to interpret observations.

The reviewer points to an important distinction between what the observations and model each indicate about subglacial drainage beneath Thwaites Glacier. Low specularity does not necessarily indicate channelization, but this is a hypothesis that our modeling seeks to test. As mentioned above, low specularity indicates a subglacial drainage system that is below capacity for any number of reasons, and our modeling is designed to determine the reason for the low specularity near the terminus. By themselves, the only conclusion that can be drawn independently from the specularity observations is the pooling of water in regions of high specularity content. Beyond this, the model must be used to test hypotheses of drainage configurations, discharge, water pressure, etc. (more detail given in response to above paragraph). Please see section 3.1.1 for a description of which model parameters yielded runs that were compatible with observations.

A sentence was added at Lines 211–213 stating the necessity of numerical modeling to explain the origin of the weakly specular regions beneath Thwaites Glacier. This is further elaborated on in 552–554 in Appendix B.

To finish, I wonder how a less-discrete compatibility criteria would compare to this method. For example, if you were to use the L2 error between the normalised S and  $R_{Wt}$  fields. I imagine that this would resemble criteria 2, but would not require choosing critical thresholds. I am not suggesting the authors include this at all in their paper, I am just making a general comment.

This is an interesting suggestion for a comparison method, although physical complications could create unreliable comparisons in regions of low specularity and low  $R_{wt}$ . High specularity content and high  $R_{wt}$  both unequivocally represent broad, flat areas of pooled water, yet the two are governed by independent processes and likely do not covary when their values are low. This makes comparison of the two difficult, and an L2 error unlikely to work as a comparison method. Comparisons between the two should instead rely on spatial point patterns (such as our binary masks) that map where specularity content and  $R_{wt}$  are high. Unfortunately, this method does require choosing critical thresholds of what is considered "high" for each quantity. We address this problem by creating a population of masks for each variable, each using a different critical threshold within a reasonable range, and comparing all 66 combinations of specularity content and  $R_{wt}$  masks. Data-compatible runs only have to match one mask combination, which makes our comparison less sensitive to our choices of critical thresholds. A more in-depth discussion of this comparison method can be found in our response to RC1.

Lines 227–230 were added to the text to briefly explain why typical correlation methods or an L2 error are unlikely to work well as a comparison method. A more detailed explanation of our comparison method is then provided in Appendix B.

In summary, the authors present a sensible methodology for making inference from some observed data (in conjunction with other physical constraints). As a consequence, they suggest that there may be significant channelisation beneath Thwaites Glacier. The existence of stable channels beneath Thwaites will have significant impact on the future of the glacier.

Overall I thought this paper was well written, and the conclusions well reasoned.

**Specific comments**

1) Given the importance of correlation as a measure of similarity I think it is important to say exactly how the correlation between masks of  $R_{Wt}$  and S is calculated.

Correlation is computed using:

$$r = \frac{\sum_m \sum_n (A_{mn} - \bar{A})(B_{mn} - \bar{B})}{\sqrt{(\sum_m \sum_n (A_{mn} - \bar{A})^2)(\sum_m \sum_n (B_{mn} - \bar{B})^2)}}$$

Where A and B are the specularity and  $R_{wt}$  masks, respectively, and the overbar denotes an average. This equation can be added into the manuscript.

This equation was included in Appendix B as Equation B1 (Line 559)

2) In 3.1.1 you state how many runs remained after eliminating unsteady runs that don't satisfy criteria 1 and 2. However, for the remainder of the paper you only use runs data compatible runs, which also have sufficient water pressure. Did this additional criteria eliminate any of the 20/14 steady state runs that satisfy the comparison criteria? If so, it would be interesting to

state here how many of your runs in total were data compatible. If not, is this condition (water pressure) at all necessary to include?

No data-compatible runs had water pressures below 90%, so this criterion can be removed from this section.

The >90% flotation criterion was removed as a necessity for data-compatibility, as no datacompatible runs were below 91% flotation, making the criterion redundant. Instead, we only mention it in Lines 159 and 170 as an indicator of when to stop expanding conductivity parameter space for each set of bed roughness parameters.

3) Section 2.4 was a bit unclear. It wasn't until the start of 3.1.1 that I knew how the criteria were applied. I think the second half of 2.4 should be re-thought to clarify the methodology. Particularly because I think the methodology is key to this paper. I think it is important to highlight that for each simulation there is 66 specularity— $R_{Wt}$  combinations to compare and that if one of these combinations satisfy the criteria then the simulation is deemed realistic. I think Lines 226 and 227 say what needs to be said at the end of 2.4 (rather than line 215 which is too vague).

We acknowledge this section is unclear and will move lines 226–227 up to section 2.4 to increase clarity.

Lines 226–227 in the original text was moved up to Section 2.4 to clarify how our comparison criteria are applied. This sentence is now at Lines 250–251 in the revised manuscript, and this is elaborated on with a new sentence at Lines 251–253.

4) Did the specularity—*R*Wt combinations suggest any particular, consistent values of critical S or *R*Wt ? Presumable SCrt is an important parameter by which we can interpret specularity results?

All 66 combinations of S and  $R_{wt}$  thresholds yielded successful comparisons for some sets of parameters, although which combinations yielded successful matches varied with conductivity and roughness parameters. Across all runs, comparisons were more successful with higher  $R_{wt}$  thresholds, with critical  $R_{wt}$  values of 0.99 or 1.0 accounting for 60% of all matches. Conversely, match success rate was not sensitive to the choice of specularity threshold, and each threshold value was responsible for 7–10% of successful matches.

A new paragraph was added to Section 3.1.1 at Lines 265–268 with the above information.

**Technical corrections**

In equation 6: is  $q_c$  a scalar value? If so, it would be clearer if it was not in bold. And if so, how is it calculated? Discharge is a vector so it isn't clear what the "discharge in the distributed system within a distance  $l_c$ " means. Presumable there it involves some integration of a dot product

taken with respect to a direction. (If it is a vector, the absolute value of a vector should probably be clarified to mean the L2 norm of the vector.) As it stands, more information about  $q_c$  is required to understand equation (6).

The authors agree  $q_c$  should be treated as a scalar, as it is only defined in one dimension. In responding to this comment, we also noticed a typo in the original equation. Equation 6 should read:

$$\Xi = \left| Q \; \frac{\partial \phi}{\partial s} \right| + \left| l_c q_c \frac{\partial \phi}{\partial s} \right|$$

This will be changed in the next version of the manuscript.

Equation 6 has been fixed (Line 90) and  $q_c$  is now treated as a scalar.

Line 117 : surfface -> surface

This will be fixed in a subsequent draft of the manuscript

This has been fixed in the manuscript (Line 120).

Figure 2 is first referenced on line 206. The caption for Figure 2 refers to FSS (flux steady- state) before this abbreviation is introduced in the text. It is not until line 228 that that FSS is defined to mean flux steady-state. Maybe just say flux steady-state rather than FSS in the caption of figure 2?

This will be fixed in a subsequent draft of the manuscript.

The caption of Figure 3 (formerly Figure 2) was changed to read "flux steady-state" instead of "FSS".

**REFERENCES:**

Schroeder, D. M., Blankenship, D. D., and Young, D. A.: Evidence for a water system transition beneath Thwaites Glacier, West Antarctica., Proceedings of the National Academy of Sciences of the United States of America, 110, 12225–8, https://doi.org/10.1073/pnas.1302828110, 2013

**Additional Minor Edits by Authors:**

Line 259: "Of our channel-enabled runs" was changed to "Of our 113 channel-enabled runs".

Line 260: 14 flux steady-state runs changed to 13 flux steady-state runs. This was a typo.

Figure 4: The middle contour line was mislabeled, and has been changed to 1.5 m/s.

Bibliography: Addition of new references used when making edits to the manuscript, as well as editing formatting errors and typos.

---

## Referee Report (RR1)

Referee comment on "Persistent, Extensive Channelized Drainage Modeled Beneath Thwaites Glacier, West Antarctica"

I appreciate the detail response of the authors. They have adequately addressed my questions. The addition of the Appendix further enhances the strength of the manuscript, especially the addition of Fig, B1 adds further clarity to the analysis. I must say the authors did good job to improve the readability of the manuscript, but I am sorry to say that it is not sufficient yet. Therefore, I still have some concerns and all of my concerns are basically around the comments that I made last time. Although, I am still not sure how robust this analysis is, but it seems convincing considering the limitations.

- 1. Flux-steady state: Where from the assumption comes that "For flux steady-state runs, meltwater production above each transect must equal the total discharge across the transect within 0.5%." Do you take it from some other paper or it's your assumption? Please provide detail justification.
- 2. The authors have taken an unconventional way to support the channelization beneath Thwaites glacier by comparing specularity content with model derived Rwt to. The approach is commendable, but simultaneously, being it bit unconventional (and ad hoc, as also author mentioned), a step-by-step clarification and discussion is required. I understand the authors have taken very conservative approach to match specularity with model, but the approach is still bit ad hoc and not easy to understand. The comparison conducted between Specularity and Rwt is extensive, and I appreciate it, but it is not clearly reflecting in text. I would strongly recommend the authors to extend this section. Also, include the cdf figure in the appendix (provided in response letter). Explain, what possible Rwt values you may get from the model run and then justify your choice of threshold. Without this progression, it is very confusing. If required, split section 2.4 into subsections and add more figures in appendix.
- 3. Another confusing part of the manuscript is absolute values of specularity and relative specularity. This is not clear what do you mean by these two specularity term. In addition, in the response letter, figure RC1, you showed a figure with specularity value up to 1. Is this absolute or relative specularity. In either way, this needs to be clarified. This is one of the important parts of the paper, but this severely lacks clarity.
- 4. The discussion part of the manuscript is very rich with various geophysical aspect, but except the discussion about Specularity and Rwt. I think there is ample opportunity to discuss about Rwt values, what can be its possible physical meaning, what they infer, whether this can be compared with other data, etc. The reason I am emphasizing on this is because this is a new approach and that's why it needs detail analysis and discussion. So far, it is vaguely convincing and incomprehensible.
- 5. One technical question: What do you change in the model to disable channelization? Can you provide a distributed Rwt figure for distributed only run, like Fig. 3c? Do

you use same threshold of Rwt of 0.95-1 for distributed only run for comparison with specularity content? It is not clear there what threshold is used (Figure 7).

I do not have any further comments on the rest of the manuscript. The paper has all the things to get it published, but only after the above concerns are adequately addressed and incorporated in the manuscript.

---

## Author Response (AR2)

Dear Dr. Kang and Reviewers,

We are very appreciative of your time in providing thorough comments and feedback for improving our manuscript. We are encouraged that both reviewers thought the manuscript was either ready for publication or close to the point of publication. We have made minor revisions to the manuscript in response to our anonymous reviewer's suggestions. The most significant changes are a new paragraph in Section 4.1, as well as the inclusion of two additional figures in Appendix B: 1) a flow chart of our methodology for comparing specularity content with model results (Figure B1), and 2) a plot of the cumulative density function of our specularity data (Figure B2). We also made minor wording changes to improve the manuscript's readability and inserted a few additional statistics regarding our choice of $R_{wt}$ thresholds to improve the credibility of our methods. In cases where we felt a change to the manuscript was not warranted, we provided our justification for our choice. We hope these minor revisions satisfy the reviewer's concerns.

In this document, reviews are given as blue italicized text, and our responses and descriptions of the revisions made are provided in black text. References to specific lines and figures in the manuscript refer to the revised text. All changes are highlighted in the 'track changes' version of the manuscript. Again, we thank you for your time and careful reading of our manuscript.

Sincerely,

Alex Hager

**Referee comment on "Persistent, Extensive Channelized Drainage Modeled Beneath Thwaites Glacier, West Antarctica"**

*I appreciate the detail response of the authors. They have adequately addressed my questions. The addition of the Appendix further enhances the strength of the manuscript, especially the addition of Fig, B1 adds further clarity to the analysis. I must say the authors did good job to improve the readability of the manuscript, but I am sorry to say that it is not sufficient yet. Therefore, I still have some concerns and all of my concerns are basically around the comments that I made last time. Although, I am still not sure how robust this analysis is, but it seems convincing considering the limitations.*

1. *Flux-steady state: Where from the assumption comes that "For flux steady-state runs, meltwater production above each transect must equal the total discharge across the transect within 0.5%." Do you take it from some other paper or it's your assumption? Please provide detail justification.*

   The flux steady-state criterion is a metric used for this paper and is not taken from previous studies. However, this is already a more robust and quantitative criterion than what is used in most other subglacial hydrology modeling studies, which either do not define a steady-state criterion or do so qualitatively. Ideally, up-glacier meltwater production would exactly equal discharge across each transect; however, this is never actually the case. Therefore, a cutoff point must be defined to establish when a model run can be categorized as being at a steady state. We include our quantitative criterion in the

text to be as thorough and open about our methods as possible, but this is beyond the norm in subglacial hydrology modeling papers. Appendix A includes an analysis of how much our choice of steady-state criterion affects our results, and the current revision includes an addition clause at lines 561–562 to emphasize the negligible uncertainty it contributes.

2. *The authors have taken an unconventional way to support the channelization beneath Thwaites glacier by comparing specularity content with model derived Rwt to. The approach is commendable, but simultaneously, being it bit unconventional (and ad hoc, as also author mentioned), a step-by-step clarification and discussion is required. I understand the authors have taken very conservative approach to match specularity with model, but the approach is still bit ad hoc and not easy to understand. The comparison conducted between Specularity and Rwt is extensive, and I appreciate it, but it is not clearly reflecting in text. I would strongly recommend the authors to extend this section. Also, include the cdf figure in the appendix (provided in response letter). Explain, what possible Rwt values you may get from the model run and then justify your choice of threshold. Without this progression, it is very confusing. If required, split section 2.4 into subsections and add more figures in appendix.*

A flow chart illustrating the step-by-step process for comparing specularity content with Rwt was added to Appendix B (Figure B1), along with the CDF figure described by the reviewer (Figure B2). If the steps outlined in Section 2.4 and Appendix B are still unclear, we kindly ask the reviewer to provide specific items that need more clarification.

As defined by Equation 12, values of Rwt can be any positive quantity, but should rarely exceed 1, as this would require the distributed system to be above its capacity and effective pressures to be zero. Rwt must be near or above 1 to resemble a flat, highly specular surface, although there should be no difference in specularity between Rwt = 1 and Rwt > 1, because both would create a perfectly flat, specular surface. Thus, the only discretion we exert in choosing Rwt thresholds is to determine a lower bound. Comparison success rates exponentially increased with higher Rwt, with only 4% of successful matches occurring with a Rwt threshold of 0.95 (as opposed to a success rate of 34% for a threshold of 1.0). The few runs that had successful matches with a Rwt threshold of 0.95 also had successful matches using higher Rwt thresholds, indicating this choice of lower bound does not influence our results. This information was added to the results at lines 267–270 to improve the credibility of our methods.

3. *Another confusing part of the manuscript is absolute values of specularity and relative specularity. This is not clear what do you mean by these two specularity term. In addition, in the response letter, figure RC1, you showed a figure with specularity value up to 1. Is this absolute or relative specularity. In either way, this needs to be clarified. This is one of the important parts of the paper, but this severely lacks clarity.*

Absolute values of specularity content depend on survey-specific factors, such as survey geometry and radar processing, and thus specularity should only be compared relative to other data within a specific survey. Therefore, values that may qualify as high specularity

in a survey of Glacier A may not be deemed high specularity in a separate survey of Glacier B. There is no difference in the values of absolute and relative specularity, just the classification of high or low specularity. We acknowledge the wording in this section is confusing, and we made minor wording changes to lines 235–238 to clarify this distinction. The CDF figure mentioned above was used to illustrate our determination of "high" specularity content within our dataset, and is now Figure B2 in the revised text.

4. *The discussion part of the manuscript is very rich with various geophysical aspect, but except the discussion about Specularity and Rwt. I think there is ample opportunity to discuss about Rwt values, what can be its possible physical meaning, what they infer, whether this can be compared with other data, etc. The reason I am emphasizing on this is because this is a new approach and that's why it needs detail analysis and discussion. So far, it is vaguely convincing and incomprehensible.*

Following the reviewer's suggestion, we inserted a paragraph into the discussion at lines 264–372 connecting the regions of high Rwt and specularity to our reconciled framework of Thwaites Glacier subglacial hydrology. Here, we interpret these regions as the "pooling of broad, flat water bodies in a distributed system at or near its capacity." This agrees with the physical definition of Rwt provided in Figure 2, lines 219–221, lines 225––227, and in the Figure 7 caption. If this is not sufficient to meet the reviewer's concerns, we kindly ask the reviewer to provide specific lines from the manuscript where it could be improved and an explanation of what is unclear.

Rwt was developed here specifically as a corollary to radar specularity content, and we refrain from speculating what other observational data it may be compared to, as this is outside the scope of the paper.

5. *One technical question: What do you change in the model to disable channelization? Can you provide a distributed Rwt figure for distributed only run, like Fig. 3c? Do you use same threshold of Rwt of 0.95-1 for distributed only run for comparison with specularity content? It is not clear there what threshold is used (Figure 7).*

The model is written with an option to disable channelization before initiating the run. Should this option be enacted, the model proceeds without defining channels between cells and Equations 4–6 are ignored.

Comparison between specularity content and Rwt is consistent for channel-enabled and distributed-only runs, including the thresholds used for defining the Rwt and specularity masks. Following the reviewer's suggestion, we have altered Figure 7 to include maps of Rwt (panels a–c), as well as the corresponding Rwt masks (panels d–f). To encompass the full range of Rwt patterns in distributed-only runs, we chose model runs for this figure with three different $k_q$ values and masks with three different Rwt thresholds. The values of $k_q$ and Rwt used in each is provided in panels d–f. In general, Rwt in distributed-only runs resembled one of these three patterns, and none resembled the data-compatible Rwt pattern shown in Figure 3.

*I do not have any further comments on the rest of the manuscript. The paper has all the things to get it published, but only after the above concerns are adequately addressed and incorporated in the manuscript.*

---

## Author Response (AR3)

**Author's Response**

Thanks for the work in adding more details to the method with clarification. With that, I believe, the manuscript has much better readability, with very interesting results and detail discussions that were already there. I do not have any further comments.

Two minor corrections need to be made.

P 16, Line 328-335: This is the caption of figure 3, somehow appeared here.

Caption removed from text

Fig B1 caption - 'step-by-step'

Typo fixed

Fig B2 caption - '94th'

Typo fixed

With this, in my opinion, the manuscript is ready for publication.